# UNCERTAINTY MODELING IN GRAPH NEURAL NETWORKS VIA STOCHASTIC DIFFERENTIAL EQUATIONS

**Richard Bergna**[1]\*, **Sergio Calvo-Ordoñez**[2,3], **Felix L. Opolka**[4],
**Pietro Liò**[4], **Jose Miguel Hernandez-Lobato**[1]
[1]Department of Engineering, University of Cambridge
[2]Mathematical Institute, University of Oxford
[3]Oxford-Man Institute of Quantitative Finance, University of Oxford
[4]Department of Computer Science and Technology, University of Cambridge

## ABSTRACT

We propose a novel Stochastic Differential Equation (SDE) framework to address the problem of learning uncertainty-aware representations for graph-structured data. While Graph Neural Ordinary Differential Equations (GNODEs) have shown promise in learning node representations, they lack the ability to quantify uncertainty. To address this, we introduce Latent Graph Neural Stochastic Differential Equations (LGNSDE), which enhance GNODE by embedding randomness through a Bayesian prior-posterior mechanism for epistemic uncertainty and Brownian motion for aleatoric uncertainty. By leveraging the existence and uniqueness of solutions to graph-based SDEs, we prove that the variance of the latent space bounds the variance of model outputs, thereby providing theoretically sensible guarantees for the uncertainty estimates. Furthermore, we show mathematically that LGNSDEs are robust to small perturbations in the input, maintaining stability over time. Empirical results across several benchmarks demonstrate that our framework is competitive in out-of-distribution detection, robustness to noise, and active learning, underscoring the ability of LGNSDEs to quantify uncertainty reliably.

## 1 INTRODUCTION

Before the widespread of neural networks and the boom in modern machine learning, complex systems in various scientific fields were predominantly modelled using differential equations. Stochastic Differential Equations (SDEs) were the standard approach to incorporating randomness. These methods were foundational across disciplines such as physics, finance, and computational biology (Hoops et al., 2016; Quach et al., 2007; Mandelzweig & Tabakin, 2001; Arroyo et al., 2024; Moreno-Pino et al., 2024; Cardelli, 2008; Buckdahn et al., 2011; Cvijovic et al., 2014).

In recent years, Graph Neural Networks (GNNs) have become the standard for graph-structured data due to their ability to capture relationships between nodes. They are widely used in social network analysis, molecular biology, and recommendation systems. However, traditional GNNs cannot reliably quantify uncertainty. Both aleatoric (inherent randomness in the data) and epistemic (model uncertainty due to limited knowledge) are essential for decision-making, risk assessment, and resource allocation, making GNNs less applicable in critical applications.

To address this gap, we propose Latent Graph Neural Stochastic Differential Equations (LGNSDE), a method that perturbs features during both the training and testing phases using Brownian motion noise, allowing for handling noise and aleatoric uncertainty. We assume a prior SDE on the latent space and learn a posterior SDE using a GNN as the drift function. This Bayesian approach to the latent space allows us to quantify epistemic uncertainty. As a result, our model can capture and quantify both epistemic and aleatoric uncertainties. More specifically, our contributions are as follows:

- We introduce a novel model class combining SDE robustness with GNN flexibility for handling complex graph-structured data, which quantifies both epistemic and aleatoric uncertainties.

- We provide theoretical guarantees demonstrating our model's ability to provide meaningful uncertainty estimates and its robustness to perturbations in the inputs.

---

\*Corresponding Author. Email: `rsb63@cam.ac.uk`

- We empirically show that Latent GNSDEs demonstrate exceptional performance in uncertainty quantification, outperforming Bayesian GCNs (Hasanzadeh et al., 2020), GCN ensembles (Lin et al., 2022) and Graph Gaussian Process (Borovitskiy et al., 2021).

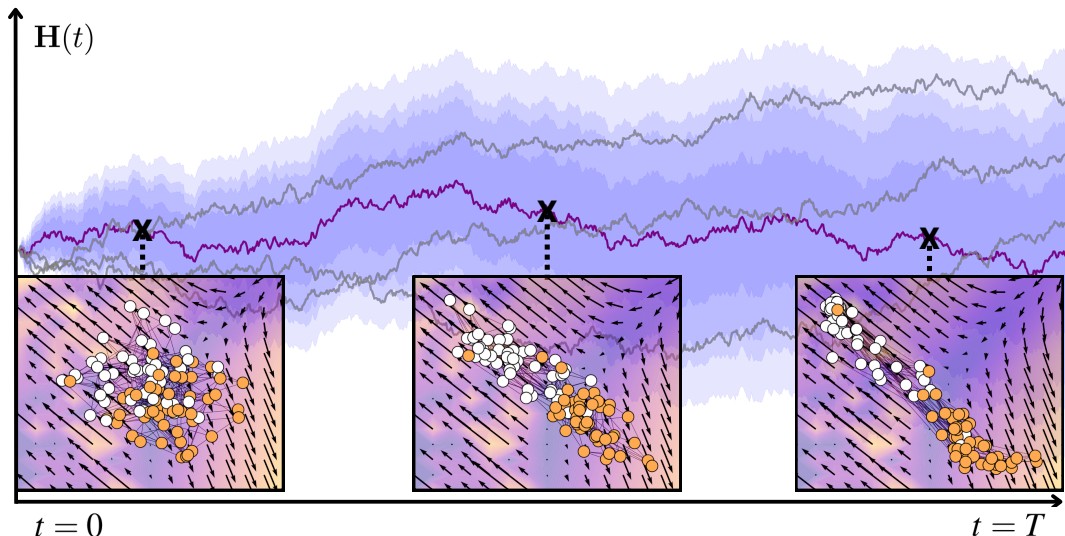

Figure 1: The diagram shows the evolution of one of the nodes of the input graph in latent space, $\mathbf{H}(t)$, through an SDE, with sample paths (purple) and confidence bands representing variance. At three timesteps, we visualize graph embeddings, where nodes (white and orange) become more separable over time due to the influence of the vector field. The inset axes represent latent dimensions, while the purple and yellow background highlights the magnitude and direction of the vector field guiding the latent dynamics.

## 2 BACKGROUND

**Graph Neural Ordinary Differential Equations (GNODE).** Introduced by Poli et al. (2019), GNODEs extend Graph ResNets by modelling node representations continuously over time. In a typical Graph ResNet, the node features evolve according to the update rule

$$\mathbf{H}(t+1) = \mathbf{H}(t) + \mathbf{F}_{\mathcal{G}}(\mathbf{H}(t), t, \theta),$$

where $t$ represents the layer index, $\mathbf{H}(t)$ and $\mathbf{H}(t+1)$ are the input and output node embeddings, and $\mathcal{G} = (\mathcal{V}, \mathcal{E})$ is a graph with node set $\mathcal{V}$ and edge set $\mathcal{E}$. The function $f_\theta$, parameterized by $\theta$, defines the transformation applied to node features. Here, the input features are $\mathbf{X}_{\text{in}} = \mathbf{H}(0)$, and the final node embeddings $\mathbf{H}(T)$ yield the model's output predictions, denoted $\hat{\mathbf{Y}}$. Now, consider an update with a small time step $c \in \mathbb{R}$

$$\mathbf{H}(t+c) = \mathbf{H}(t) + c \cdot \mathbf{F}_{\mathcal{G}}(\mathbf{H}(t), t, \theta),$$

which leads to the continuous limit as $c \to 0$

$$\frac{\mathrm{d}\mathbf{H}(t)}{\mathrm{d}t} = \mathbf{F}_{\mathcal{G}}(\mathbf{H}(t), t, \theta).$$

This differential equation represents the continuous evolution of node features over time, transforming the discrete depth of layers into a continuous variable $t$. The solution to this equation is given by

$$\mathbf{H}(t) = \mathbf{H}(0) + \int_0^t \mathbf{F}_{\mathcal{G}}(\mathbf{H}(u), u, \theta) \, \mathrm{d}u.$$

In practice, $\mathbf{F}_{\mathcal{G}}$ is modelled by a neural network, and $t$ operates as a continuous depth parameter. The solution to the integral is approximated numerically, making GNODE a continuous-depth analogue of graph residual networks.

## 3 METHODOLOGY

Inspired by Graph Neural ODEs (Poli et al., 2019) and Latent SDEs (Li et al., 2020), we now introduce our model: Latent Graph Neural SDEs − LGNSDEs (Figure 1), which use SDEs to define prior and approximate posterior stochastic trajectories for $\mathbf{H}(t)$ (Xu et al., 2022). Furthermore, LGNSDEs can be viewed as the continuous representations of existing discrete architectures (A.4).

### 3.1 MODEL DEFINITION

LGNSDEs are designed to capture the stochastic latent evolution of $\mathbf{H}(t)$ on graph-structured data. We use an Ornstein-Uhlenbeck (OU) prior process, represented by

$$d\mathbf{H}(t) = \mathbf{F}_{\mathcal{G}}(\mathbf{H}(t), t)\, dt + \mathbf{G}_{\mathcal{G}}(\mathbf{H}(t), t)\, d\mathbf{W}(t),$$

where we set the drift and diffusion functions, $\mathbf{F}_{\mathcal{G}}$ and $\mathbf{G}_{\mathcal{G}}$, to constants and consider them hyperparameters. Moreover, $d\mathbf{W}(t)$ is a Wiener process. The approximate posterior is defined as

$$d\mathbf{H}(t) = \mathbf{F}_{\mathcal{G}}(\mathbf{H}(t), t, \phi)\, dt + \mathbf{G}_{\mathcal{G}}(\mathbf{H}(t), t)\, d\mathbf{W}(t), \tag{1}$$

where $\mathbf{F}_{\mathcal{G}}$ is now parameterized by a GCN with $\phi$ representing the learned weights of the neural network. The drift function mainly determines the dynamics of the evolution of the latent state, while the diffusion term $\mathbf{G}_{\mathcal{G}}(\mathbf{H}(t))\, d\mathbf{W}(t)$ introduces stochastic elements. With the need to keep the Kullback-Leibler (KL) divergence bounded, we set the diffusion functions, $\mathbf{G}_{\mathcal{G}}$, of the prior and posterior to be the same [Calvo-Ordonez et al. 2024; Archambeau et al. 2007].

Let $\mathbf{Y}$ be a collection of target variables, e.g., class labels, for some of the graph nodes. Given $\mathbf{Y}$ we train our model with variational inference, with the ELBO computed as

$$\mathcal{L}_{\text{ELBO}}(\phi) = \mathbb{E}\left[\log p(\mathbf{Y}|\mathbf{H}(t)) - \int_0^t \frac{1}{2}\, \|v(\mathbf{H}(u), \phi, \theta, \mathcal{G})\|_2^2\, du\right],$$

where the expectation is approximated over trajectories of $\mathbf{H}(t)$ sampled from the approximate posterior SDE, and $v = \mathbf{G}_{\mathcal{G}}(\mathbf{H}(t))^{-1}[\mathbf{F}_{\mathcal{G},\phi}(\mathbf{H}(u), u) - \mathbf{F}_{\mathcal{G},\theta}(\mathbf{H}(u), u)]$.

We sample $\mathbf{H}(t)$ by integrating the SDE in Eq. 1. The analytical solution is

$$\mathbf{H}(t) = \mathbf{H}(0) + \int_0^t \mathbf{F}_{\mathcal{G},\phi}(\mathbf{H}(u), u)\, du + \int_0^t \mathbf{G}_{\mathcal{G}}(\mathbf{H}(u), u)\, d\mathbf{W}(u),$$

where $\mathbf{H}(0)$ are the node-wise features $\mathbf{X}_{\text{in}}$ in the graph $\mathcal{G}$. We numerically solve this integral with a standard Stochastic Runge-Kutta method (Rößler, 2010). We then use a Monte Carlo approximation to get the expectation of $\mathbf{H}(t)$ and approximate the posterior predictive distribution as

$$p(\mathbf{Y}^*|\mathcal{G}, \mathbf{X}_{\text{in}}, \mathbf{Y}) \approx \frac{1}{N} \sum_{n=1}^{N} p\left(\mathbf{Y}^*|\mathbf{H}_n(t), \mathcal{G}\right),$$

where $\mathbf{H}_1(t_1), \ldots, \mathbf{H}_N(t)$ are samples drawn from the approximation to $p(\mathbf{H}(t)|\mathbf{Y}, \mathbf{X}_{\text{in}}, \mathcal{G})$.

Following Poli et al. (2019), we use a similar encoder-decoder setup. Our encoding focuses solely on the features of individual nodes, while the graph structure remains unchanged. Finally, we remark that the memory complexity when using the stochastic adjoint sensitivity method is $\mathcal{O}(1)$ and the time complexity is $\mathcal{O}(L \log L(|\mathcal{E}|d + |\mathcal{V}|d))$, where $L$ is the number of SDE solver steps, $\mathcal{E}$ is the number of edges in the graph, $\mathcal{V}$ is the number of nodes, and $d$ is the dimension of the features (see Appendix C.3). For a runtime comparison with other models see 12, and 15.

Note that in our framework, model depth is inherently tied to the evolution of the latent space, where depth is determined by the number of layers corresponding to the sampling steps of the SDE solver. As the SDE solver dictates the number of steps, it effectively controls the number of layers in the model. Thereby the SDE solver chooses the number of layers, dynamically changing the model's complexity based on task difficulty. For more complex tasks, the solver will generate additional steps (layers), while simpler tasks will require fewer layers. Exploring optimal adaptive SDE solvers will remain part of future work.

## 4 THEORETICAL GUARANTEES

To establish the theoretical foundations of our framework, we begin by acknowledging the existence and uniqueness results for graph-based stochastic differential equations, as proven in Lin et al. (2024). This result guarantees that, under certain assumptions (Appendix A.2), there exists a unique mild solution to the Graph Neural SDE, ensuring the well-posedness of the model's dynamics and that the solution behaves in a stable and predictable manner, i.e. small changes in the initial conditions or input data lead to small changes in the solution. We borrow this result in the following theorem:

**Theorem 1** (Lin et al. (2024)). *If $\Psi$ is a linear operator with a complete orthonormal basis set and eigenvalues $\lambda_k > 0$, the continuous operator $\mathbf{G}_\mathcal{G}$ satisfies the Lipschitz condition, and the initial node representation $\mathbf{H}_i(0)$ is square-integrable and $\mathcal{F}_0$-measurable, then there exists a unique mild solution $\mathbf{H}_i(t)$ on $[0, T]$ for any $T > 0$ and $i \in \mathcal{V}$, such that*

$$\mathbf{H}_i(t) = e^{t\Psi}\mathbf{H}_i(0) + \int_0^t e^{(t-s)\Psi}\mathbf{G}_\mathcal{G}(\mathbf{H}_i(s))d\mathbf{W}(s),$$

*where $e^{t\Psi}$ is the semigroup generated by $\Psi$. Furthermore, there exists a constant $C_T > 0$ such that*

$$\sup_{t \in [0,T]} \|\mathbf{H}_i(t)\| \leq C_T(1 + \|\mathbf{H}_i(0)\|).$$

This theorem confirms that graph-based SDEs have a well-posed solution trajectory over time that is well-behaved and bounded, ensuring that our LGNSDE model can maintain stability across varying graph structures, meaning that the solution does not exhibit erratic or unbounded behaviour even under small changes in graph structure or input features.

Leveraging this result, we proceed to present key results on the stability and robustness of our framework.

- We derive a bound that proves that our proposed models provide meaningful uncertainties.
- We demonstrate the robustness of our framework under small perturbations in the initial conditions.

By showing that the variance of the latent representation bounds the model output variance, we highlight the ability of LGNSDEs to capture and quantify inherent uncertainty in the system. The model's output is given by the trajectory of the latent representation since $\mathbf{y} = \mathbf{H}(t = T)$, therefore the uncertainty in the latent space directly influences the uncertainty in predictions. We formalize this in the following proposition:

**Proposition 1.** *Under assumptions 1-3 and given Theorem 1, there exists a unique mild[1] solution to an LGNSDE of the form*

$$d\mathbf{H}(t) = \mathbf{F}_\mathcal{G}(\mathbf{H}(t), t, \boldsymbol{\theta})\, dt + \mathbf{G}_\mathcal{G}(\mathbf{H}(t), t)\, d\mathbf{W}(t),$$

*whose variance bounds the variance of the model output $\hat{\mathbf{y}}(t)$ as:*

$$Var(\hat{\mathbf{y}}(t)) \leq L_h^2 Var(\mathbf{H}(t)),$$

*where $L_h^2$ is the Lipschitz constant of the readout layer. This ensures that the output variance is bounded by the prior variance of the latent space, providing a controlled measure of uncertainty.*

Now, by deriving explicit bounds on the deviation between the perturbed and unperturbed solutions over time, we show that the model's output remains stable.

**Proposition 2.** *Under assumptions 1-3, consider two initial conditions $\mathbf{H}_0$ and $\tilde{\mathbf{H}}_0 = \mathbf{H}_0 + \delta\mathbf{H}(0)$, where $\delta\mathbf{H}(0) \in \mathbb{R}^{n \times d}$ is a small perturbation in the initial node features with $\|\delta\mathbf{H}(0)\|_F = \epsilon$. Assume that $\mathbf{H}_0$ is taken from a compact set $\mathcal{H} \subseteq \mathbb{R}^{n \times d}$. Then, the deviation between the solutions $\mathbf{H}(t)$ and $\tilde{\mathbf{H}}(t)$ of the LGNSDE with these initial conditions remains bounded across time $t$[2], specifically*

$$\mathbb{E}[\|\mathbf{H}(t) - \tilde{\mathbf{H}}(t)\|_F] \leq \epsilon e^{(L_f + \frac{1}{2}L_g^2)t}.$$

---

[1] A mild solution to an SDE is expressed via an integral equation involving the semigroup generated by the linear operator and represents a weaker notion of the solution.

[2] Note that while the bound is exponential in $t$, in practice, the time horizon is usually constrained to a limited range, such as $t \in [0, 1]$. Within this interval, the exponential factor does not grow excessively, ensuring that the deviation between the perturbed and unperturbed solutions remains under control.

In summary, we show analytically that our framework effectively quantifies uncertainty and maintains robustness under small perturbations of the input. First, we confirm that the model's output variance is controlled and directly linked to the variance of the latent state. Second, we provide a bound on the deviation between solutions with perturbed initial conditions, ensuring stability over time. The proofs can be found in Appendix A.

## 5 EXPERIMENTS

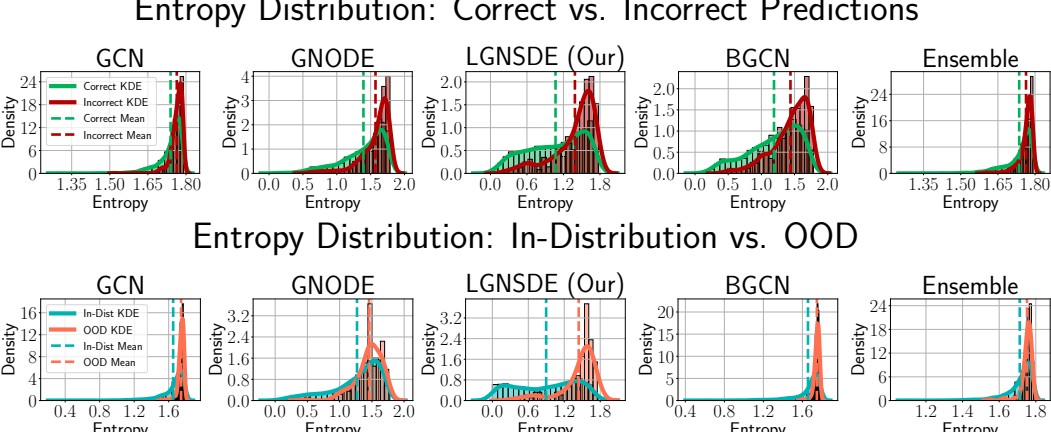

Figure 2: **Top:** Entropy distributions comparing correct and incorrect model predictions on the CORA dataset. Higher entropy is expected for incorrect predictions. **Bottom:** Entropy distributions comparing OOD samples with in-distribution samples in the CORA dataset.

| Metric | Model | Cora | Citeseer | Computers | Photo | Pubmed |
|--------|-------|------|----------|-----------|-------|--------|
| **AUROC (↑)** | GNN | 0.9654 ± 0.0050 | 0.9173 ± 0.0068 | 0.9680 ± 0.0016 | 0.9905 ± 0.0003 | 0.9006 ± 0.0139 |
| | GNODE | 0.9664 ± 0.0051 | 0.9146 ± 0.0063 | 0.9569 ± 0.0067 | 0.9885 ± 0.0007 | 0.8857 ± 0.0203 |
| | BGCN | 0.9571 ± 0.0092 | 0.9099 ± 0.0090 | 0.9421 ± 0.0097 | 0.9489 ± 0.0189 | 0.7030 ± 0.1331 |
| | ENSEMBLE | 0.9635 ± 0.0031 | 0.9181 ± 0.0062 | 0.9669 ± 0.0025 | 0.9886 ± 0.0004 | 0.8785 ± 0.0163 |
| | GRAPH GP | 0.8970 ± 0.0055 | 0.8877 ± 0.0062 | OOM | OOM | OOM |
| | **LGNSDE (Ours)** | 0.9659 ± 0.0038 | 0.9111 ± 0.0072 | 0.9691 ± 0.0032 | 0.9909 ± 0.0004 | 0.9004 ± 0.0087 |
| **AURC (↓)** | GNN | 0.0709 ± 0.0101 | 0.1626 ± 0.0109 | 0.0745 ± 0.0053 | 0.0199 ± 0.0007 | 0.1367 ± 0.0192 |
| | GNODE | 0.0628 ± 0.0095 | 0.1609 ± 0.0141 | 0.1055 ± 0.0165 | 0.0219 ± 0.0015 | 0.1427 ± 0.0214 |
| | BGCN | 0.0858 ± 0.0219 | 0.1764 ± 0.0215 | 0.1634 ± 0.0344 | 0.1218 ± 0.0577 | 0.4152 ± 0.1723 |
| | ENSEMBLE | 0.0789 ± 0.0061 | 0.1722 ± 0.0179 | 0.0877 ± 0.0037 | 0.0244 ± 0.0012 | 0.1722 ± 0.0285 |
| | GRAPH GP | 0.1869 ± 0.0084 | 0.2328 ± 0.0118 | OOM | OOM | OOM |
| | **LGNSDE (Ours)** | 0.0702 ± 0.0095 | 0.1686 ± 0.0146 | 0.0687 ± 0.0114 | 0.0186 ± 0.0007 | 0.1378 ± 0.0118 |
| **Accuracy (↑)** | GNN | 0.8105 ± 0.0173 | 0.7258 ± 0.0137 | 0.8098 ± 0.0048 | 0.9116 ± 0.0021 | 0.7570 ± 0.0229 |
| | GNODE | 0.8202 ± 0.0149 | 0.7235 ± 0.0159 | 0.7911 ± 0.0098 | 0.9053 ± 0.0032 | 0.7577 ± 0.0231 |
| | BGCN | 0.7897 ± 0.0261 | 0.7013 ± 0.0196 | 0.7114 ± 0.0333 | 0.7124 ± 0.0968 | 0.4581 ± 0.1846 |
| | ENSEMBLE | 0.8038 ± 0.0105 | 0.7108 ± 0.0166 | 0.8070 ± 0.0055 | 0.9070 ± 0.0019 | 0.7299 ± 0.0218 |
| | GRAPH GP | 0.6491 ± 0.0116 | 0.6675 ± 0.0127 | OOM | OOM | OOM |
| | **LGNSDE (Ours)** | 0.8079 ± 0.0154 | 0.7120 ± 0.0119 | 0.8247 ± 0.0103 | 0.9169 ± 0.0021 | 0.7589 ± 0.0161 |

Table 1: Performance comparison of models across five datasets (Cora, Citeseer, Computers, Photo, Pubmed) based on AUROC, AURC, and Accuracy (mean ± standard deviation). Red indicates the best-performing model, while blue indicates the second-best-performing model for each metric. Some results for the Graph GP model are unavailable due to out-of-memory (OOM) errors.

We evaluate LGNSDE on the following datasets: *Cora* (Sen et al., 2008), *CiteSeer* (Giles et al., 1998), *PubMed* (Sen et al., 2008), and the Amazon co-purchasing graphs *Computer* (McAuley et al., 2015) and *Photo* (Shchur et al., 2018). We compare its performance against GNODE (Poli et al., 2019), GCN (Kipf & Welling, 2016), Bayesian GCN (BGCN) (Hasanzadeh et al., 2020), an ensemble of GCNs (Lin et al., 2022) and Graph GPs[3] (Borovitskiy et al., 2021).

---

[3]We were unable to run the Graph GP model on certain datasets due to its high memory requirements and scalability issues, resulting in out-of-memory (OOM) errors. Consequently, we report these cases as OOM in our experimental results tables.

In conducting our experiments, we used the setup outlined in Shchur et al. (2018). This involved using 20 random weight initializations for datasets with fixed Planetoid splits and implementing 100 random splits for other datasets. The hyperparameters that achieved the highest validation accuracy were chosen, and their performance was evaluated on a test set. For further details on our hyperparameter grid search refer to Appendix B.

## 5.1 STANDARD SETTING

The results in Table 1 demonstrate that LGNSDE consistently ranks as either the best or second-best model across most datasets in terms of Micro-AUROC (Area Under the Receiver Operating Characteristic), AURC (Area Under the Risk Coverage), and accuracy. This indicates that LGNSDE effectively handles model uncertainty, successfully distinguishing between classes (AUROC), maintaining low risk while ensuring confident predictions (AURC), and delivering high accuracy.

The top of Figure 2 shows the entropy distributions of the models for correct and incorrect predictions. Note that most models display similar mean entropy for both correct and incorrect predictions. Notably, our model stands out with the largest difference in entropy, with incorrect predictions having 35% more entropy (more uncertainty – see Subsection 5.2) compared to correct predictions, a larger gap than observed in other models.

## 5.2 OUT OF DISTRIBUTION DETECTION

| Metric | Model | Cora | Citeseer | Computers | Photo | Pubmed |
|--------|-------|------|----------|-----------|-------|--------|
| **AUROC (↑)** | GNN | 0.7063 ± 0.0569 | 0.7937 ± 0.0366 | 0.7796 ± 0.0271 | 0.8578 ± 0.0136 | 0.6127 ± 0.0351 |
| | GNODE | 0.7398 ± 0.0677 | 0.7828 ± 0.0465 | 0.7753 ± 0.0795 | 0.8473 ± 0.0158 | 0.5813 ± 0.0242 |
| | BGCN | 0.7193 ± 0.0947 | 0.8287 ± 0.0377 | 0.7914 ± 0.1234 | 0.7910 ± 0.0464 | 0.5310 ± 0.0472 |
| | ENSEMBLE | 0.7031 ± 0.0696 | 0.8190 ± 0.0375 | 0.8292 ± 0.0338 | 0.8352 ± 0.0059 | 0.6130 ± 0.0311 |
| | **LGNSDE (Ours)** | 0.7614 ± 0.0804 | 0.8258 ± 0.0418 | 0.7994 ± 0.0238 | 0.8707 ± 0.0099 | 0.6204 ± 0.0162 |
| **AURC (↓)** | GNN | 0.0220 ± 0.0049 | 0.0527 ± 0.0075 | 0.0072 ± 0.0013 | 0.0076 ± 0.0006 | 0.3227 ± 0.0266 |
| | GNODE | 0.0184 ± 0.0053 | 0.0545 ± 0.0110 | 0.0070 ± 0.0029 | 0.0097 ± 0.0015 | 0.3357 ± 0.0309 |
| | BGCN | 0.0208 ± 0.0091 | 0.0458 ± 0.0071 | 0.0064 ± 0.0047 | 0.0108 ± 0.0034 | 0.3714 ± 0.0317 |
| | ENSEMBLE | 0.0215 ± 0.0061 | 0.0487 ± 0.0072 | 0.0041 ± 0.0011 | 0.0081 ± 0.0003 | 0.3277 ± 0.0265 |
| | **LGNSDE (Ours)** | 0.0168 ± 0.0070 | 0.0479 ± 0.0109 | 0.0061 ± 0.0011 | 0.0068 ± 0.0008 | 0.3205 ± 0.0135 |
| **Accuracy (↑)** | GNN | 0.9470 ± 0.0004 | 0.8614 ± 0.0071 | 0.9788 ± 0.0000 | 0.9558 ± 0.0002 | 0.6180 ± 0.0155 |
| | GNODE | 0.9469 ± 0.0002 | 0.8603 ± 0.0086 | 0.9788 ± 0.0004 | 0.9557 ± 0.0000 | 0.6084 ± 0.0120 |
| | BGCN | 0.9472 ± 0.0004 | 0.8711 ± 0.0133 | 0.9797 ± 0.0010 | 0.9558 ± 0.0002 | 0.6039 ± 0.0074 |
| | ENSEMBLE | 0.9470 ± 0.0003 | 0.8699 ± 0.0113 | 0.9788 ± 0.0001 | 0.9560 ± 0.0001 | 0.6216 ± 0.0112 |
| | **LGNSDE (Ours)** | 0.9471 ± 0.0003 | 0.8729 ± 0.0108 | 0.9788 ± 0.0000 | 0.9560 ± 0.0002 | 0.6243 ± 0.0094 |

Table 2: Performance comparison of models for OOD detection across five datasets (Cora, Citeseer, Computers, Photo, Pubmed). Metrics reported are AUROC, AURC, and Accuracy (mean ± standard deviation). Red indicates the best-performing model, while blue indicates the second-best-performing model for each metric.

We evaluate the models' ability to detect out-of-distribution (OOD) data by training them with one class left out of the dataset. This introduces an additional class in the validation and test sets that the models have not encountered during training. The goal is to determine if the models can identify this class as OOD. We analyze the entropy

$$H(\hat{y}|\mathbf{X}_i) = -\sum_{c=1}^{C} p(\hat{y} = c|\mathbf{X}_i) \log p(\hat{y}|\mathbf{X}_i), \tag{2}$$

where $p(\hat{y} = c|\mathbf{X}_i)$ represents the probability of input $\mathbf{X}_i$ belonging to class $c$. Entropy quantifies the uncertainty in the model's predicted probability distribution over $C$ classes for a given input $\mathbf{X}_i$.

The bottom of Figure 2 shows the test entropy distribution for in-distribution (blue) and out-of-distribution (red) data. For each test sample, predictions were made over $C - 1$ classes, excluding the left-out class. The OOD class exhibits higher entropy, indicating greater uncertainty. While most models show similar entropy distributions for both data types, our LGNSDE model achieves a clear separation, with a 50% higher mean entropy for OOD data compared to in-distribution data. Other models show less than a 10% difference between the two distributions.

Table 2 presents the AUROC and AURC scores for OOD detection across multiple datasets. AUROC evaluates the model's ability to differentiate between in-distribution and out-of-distribution

(OOD) samples, with higher scores indicating better discrimination. AURC measures the risk of misclassification as coverage increases, where lower values are preferred. LGNSDE consistently achieves the best AUROC and AURC scores across most datasets, indicating its superior performance in accurately identifying OOD samples and minimizing the risk of misclassification.

The accuracy was determined by applying an entropy-based threshold. Predictions with entropy above this threshold were classified as out-of-distribution (OOD), while those below were considered in-distribution. The optimal threshold was identified using a validation dataset, where it was selected to maximize overall classification performance.

## 5.3 NOISE PERTURBATION

| Metric | Model | Cora | Citeseer | Computers | Photo | Pubmed |
|---|---|---|---|---|---|---|
| **AUROC (↑)** | GCN | 0.9610 ± 0.0045 | 0.9096 ± 0.0056 | 0.9682 ± 0.0029 | 0.9909 ± 0.0002 | 0.8466 ± 0.0214 |
| | GNODE | 0.9649 ± 0.0054 | 0.9077 ± 0.0062 | 0.9593 ± 0.0033 | 0.9892 ± 0.0007 | 0.8727 ± 0.0183 |
| | BGCN | 0.9606 ± 0.0034 | 0.9069 ± 0.0076 | 0.9547 ± 0.0095 | 0.9845 ± 0.0028 | 0.7643 ± 0.0771 |
| | ENSEMBLE | 0.9581 ± 0.0051 | 0.9166 ± 0.0048 | 0.9701 ± 0.0011 | 0.9893 ± 0.0004 | 0.8093 ± 0.0511 |
| | GRAPH GP | 0.8979 ± 0.0050 | 0.8889 ± 0.0037 | OOM | OOM | OOM |
| | **LGNSDE (Our)** | 0.9634 ± 0.0065 | 0.9172 ± 0.0070 | 0.9698 ± 0.0015 | 0.9911 ± 0.0004 | 0.8636 ± 0.0310 |
| **AURC (↓)** | GCN | 0.0782 ± 0.0076 | 0.1755 ± 0.0079 | 0.0661 ± 0.0099 | 0.0185 ± 0.0005 | 0.2098 ± 0.0397 |
| | GNODE | 0.0625 ± 0.0109 | 0.1676 ± 0.0085 | 0.1006 ± 0.0114 | 0.0209 ± 0.0018 | 0.1654 ± 0.0260 |
| | BGCN | 0.0799 ± 0.0075 | 0.1687 ± 0.0122 | 0.1400 ± 0.0438 | 0.0359 ± 0.0059 | 0.3026 ± 0.0928 |
| | ENSEMBLE | 0.0862 ± 0.0067 | 0.1690 ± 0.0099 | 0.0718 ± 0.0037 | 0.0230 ± 0.0008 | 0.2603 ± 0.0528 |
| | GRAPH GP | 0.1827 ± 0.0081 | 0.2294 ± 0.0088 | OOM | OOM | OOM |
| | **LGNSDE (Our)** | 0.0731 ± 0.0128 | 0.1612 ± 0.0145 | 0.0642 ± 0.0059 | 0.0184 ± 0.0010 | 0.1908 ± 0.0519 |
| **Accuracy (↑)** | GCN | 0.8054 ± 0.0112 | 0.7162 ± 0.0145 | 0.8169 ± 0.0026 | 0.9139 ± 0.0013 | 0.6874 ± 0.0346 |
| | GNODE | 0.8255 ± 0.0134 | 0.7213 ± 0.0116 | 0.7907 ± 0.0126 | 0.9074 ± 0.0061 | 0.7402 ± 0.0286 |
| | BGCN | 0.7946 ± 0.0115 | 0.7034 ± 0.0223 | 0.7401 ± 0.0472 | 0.8754 ± 0.0180 | 0.5848 ± 0.0973 |
| | ENSEMBLE | 0.7916 ± 0.0156 | 0.7199 ± 0.0139 | 0.8160 ± 0.0038 | 0.9091 ± 0.0013 | 0.6390 ± 0.0577 |
| | GRAPH GP | 0.6486 ± 0.0124 | 0.6697 ± 0.0070 | OOM | OOM | OOM |
| | **LGNSDE (Our)** | 0.8101 ± 0.0179 | 0.7214 ± 0.0178 | 0.8263 ± 0.0098 | 0.9165 ± 0.0022 | 0.7100 ± 0.0422 |

Table 3: AUROC (Mean ± Std), AURC (Mean ± Std), and Accuracy (Mean ± Std) for all datasets with noise perturbations. Red denotes the best-performing model, and blue denotes the second-best-performing model.

We evaluate the models with noise added during testing to assess their robustness to input perturbations. No noise is introduced during training or validation. At test time, Gaussian noise is applied to the input feature vectors. Specifically, the noisy inputs are defined as $\mathbf{X}_{\text{new\_test}} = \mathbf{X}_{\text{test}} + 0.5 \cdot \mathcal{N}(0, \sigma)$, where $\mathcal{N}(0, \sigma)$ represents element-wise Gaussian noise with mean 0 and standard deviation $\sigma$, independently applied to each input feature. Here, $\sigma$ is computed as the standard deviation of $\mathbf{X}_{\text{test}}$, i.e., $\sigma = \text{std}(\mathbf{X}_{\text{test}})$. By adding noise scaled by 0.5 times the standard deviation of the test set, we ensure that the perturbations are proportional to the feature distribution across all datasets.

Table 3 presents the results under the noisy perturbation setting, where our model, LGNSDE, consistently ranks among the top two across all datasets and metrics (AUROC, AURC, and Accuracy). This demonstrates its robust performance under noise, frequently outperforming other models. These results also align with the guarantees provided by Proposition 2, which predict bounded deviations under input perturbations, supporting the observed robustness of LGNSDE.

## 5.4 ACTIVE LEARNING

We delve into decision-making under uncertainty in the context of an active learning setup, where the model selects its own training data. The experiments begin with the same set of observed labels as in the previous experiments (see Table 8). In each active learning round, 5 additional labels/nodes are incrementally included using an acquisition function, selected from either the validation or test dataset. After each new label is added, the models are trained for 25 epochs, and this process continues until the number of newly added labels has doubled.

Figure 3 shows the active learning experiment conducted on the Cora dataset nodes using two acquisition functions. In the right plot, labels are selected based on the highest predictive entropy, while in the left plot, they are selected randomly.

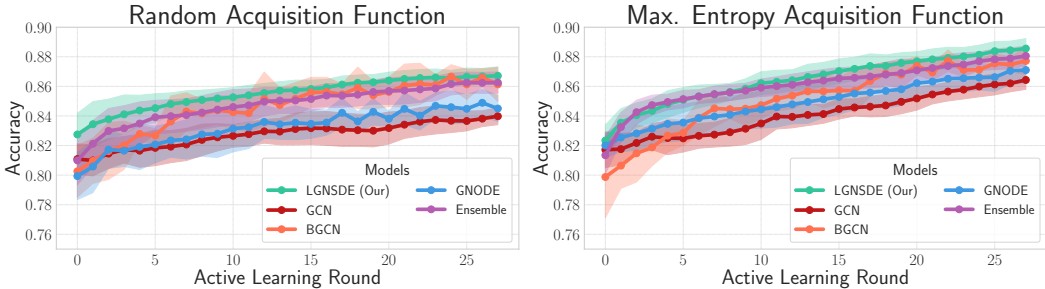

Figure 3: Active learning on the Cora dataset using two acquisition functions. The left plot shows a random selection of labels, while the right plot shows a selection based on maximum predictive entropy

| Metric | Model | Cora | Citeseer | Computers | Photo | Pubmed |
|---|---|---|---|---|---|---|
| **AUROC (↑)** | GCN | 0.9814 ± 0.0019 | 0.9447 ± 0.0017 | 0.9823 ± 0.0017 | 0.9933 ± 0.0006 | 0.9313 ± 0.0043 |
| | GNODE | 0.9732 ± 0.0030 | 0.9149 ± 0.0080 | 0.9830 ± 0.0028 | 0.9923 ± 0.0008 | 0.9108 ± 0.0166 |
| | BGCN | 0.9813 ± 0.0024 | 0.8339 ± 0.0341 | 0.9777 ± 0.0028 | 0.9896 ± 0.0023 | 0.7359 ± 0.0406 |
| | ENSEMBLE | 0.9712 ± 0.0016 | 0.9139 ± 0.0047 | 0.9830 ± 0.0013 | 0.9926 ± 0.0008 | 0.9243 ± 0.0054 |
| | GRAPH GP | 0.9434 ± 0.0050 | 0.9060 ± 0.0040 | OOM | OOM | OOM |
| | **LGNSDE (Ours)** | 0.9812 ± 0.0016 | 0.9302 ± 0.0044 | 0.9846 ± 0.0026 | 0.9933 ± 0.0004 | 0.9288 ± 0.0050 |
| **AURC (↓)** | GCN | 0.0392 ± 0.0037 | 0.1084 ± 0.0065 | 0.0442 ± 0.0020 | 0.0148 ± 0.0017 | 0.0937 ± 0.0066 |
| | GNODE | 0.0439 ± 0.0034 | 0.1396 ± 0.0159 | 0.0512 ± 0.0070 | 0.0163 ± 0.0028 | 0.1154 ± 0.0235 |
| | BGCN | 0.0399 ± 0.0049 | 0.2522 ± 0.0795 | 0.0704 ± 0.0102 | 0.0204 ± 0.0069 | 0.3574 ± 0.0695 |
| | ENSEMBLE | 0.0543 ± 0.0038 | 0.1410 ± 0.0111 | 0.0375 ± 0.0027 | 0.0143 ± 0.0013 | 0.1007 ± 0.0076 |
| | GRAPH GP | 0.1027 ± 0.0101 | 0.1975 ± 0.0108 | OOM | OOM | OOM |
| | **LGNSDE (Ours)** | 0.0387 ± 0.0033 | 0.1249 ± 0.0106 | 0.0394 ± 0.0031 | 0.0148 ± 0.0013 | 0.1030 ± 0.0094 |
| **Accuracy (↑)** | GCN | 0.8626 ± 0.0072 | 0.7720 ± 0.0051 | 0.8734 ± 0.0054 | 0.9315 ± 0.0042 | 0.8130 ± 0.0077 |
| | GNODE | 0.8450 ± 0.0088 | 0.7308 ± 0.0269 | 0.8731 ± 0.0069 | 0.9247 ± 0.0041 | 0.7888 ± 0.0207 |
| | BGCN | 0.8556 ± 0.0115 | 0.5530 ± 0.0992 | 0.8274 ± 0.0272 | 0.9049 ± 0.0150 | 0.4996 ± 0.0721 |
| | ENSEMBLE | 0.8384 ± 0.0057 | 0.7253 ± 0.0115 | 0.8844 ± 0.0055 | 0.9280 ± 0.0039 | 0.7911 ± 0.0109 |
| | GRAPH GP | 0.7727 ± 0.0114 | 0.7148 ± 0.0141 | OOM | OOM | OOM |
| | **LGNSDE (Ours)** | 0.8641 ± 0.0061 | 0.7534 ± 0.0116 | 0.8830 ± 0.0058 | 0.9318 ± 0.0012 | 0.8097 ± 0.0111 |

Table 4: Results of the active learning experiments **using a random acquisition function**. The table reports AUROC, AURC, and Accuracy (Mean ± Std) for all datasets. Red indicates the best-performing model, and blue indicates the second-best-performing model.

### 5.4.1 RANDOM ACQUISITION FUNCTION

To provide a baseline comparison with the maximum entropy acquisition function, we first evaluate the models using a random acquisition function. At each active learning round, random labels are selected to be included in the training set. Table 4 illustrates the performance of the models under this setup, showing the AUROC, AURC, and accuracy metrics across all datasets.

This setup allows us to contrast the effectiveness of random label selection with more informed selection strategies like the maximum entropy acquisition function, which we explore next.

### 5.4.2 TOTAL ENTROPY ACQUISITION FUNCTION

Now, we evaluate the models using a total entropy acquisition function, where labels with the highest uncertainty are selected at each active learning round. Table 5 shows the AUROC, AURC, and accuracy metrics for all datasets. Our model, LGNSDE, performs significantly better with the total entropy acquisition function compared to the random acquisition function (Table 4). It achieves the highest AUROC on Cora, Computers, Photo, and Pubmed, showing that it effectively selects the most informative points when uncertainty is used as a guide. In contrast, the random function led to more mixed results across these datasets. For the AURC metric, where lower is better, LGNSDE shows clear improvements with the entropy strategy, particularly on Cora, Computers, and Pubmed. This suggests that our model is better at reducing classification errors when it focuses on uncertain points, compared to the random selection method. In terms of accuracy, LGNSDE also sees gains under the total entropy function. For example, accuracy on Cora improves from 0.8641 to 0.8889, and on

| Metric | Model | Cora | Citeseer | Computers | Photo | Pubmed |
|---|---|---|---|---|---|---|
| **AUROC (↑)** | GCN | 0.9831 ± 0.0013 | 0.9434 ± 0.0053 | 0.9881 ± 0.0008 | 0.9936 ± 0.0004 | 0.9248 ± 0.0129 |
| | GNODE | 0.9789 ± 0.0027 | 0.9419 ± 0.0035 | 0.9838 ± 0.0017 | 0.9914 ± 0.0006 | 0.9113 ± 0.0140 |
| | BGCN | 0.9830 ± 0.0026 | 0.8073 ± 0.0495 | 0.9809 ± 0.0029 | 0.9915 ± 0.0014 | 0.7169 ± 0.0529 |
| | ENSEMBLE | 0.9755 ± 0.0023 | 0.9192 ± 0.0049 | 0.9867 ± 0.0005 | 0.9916 ± 0.0004 | 0.9236 ± 0.0052 |
| | GRAPH GP | 0.9520 ± 0.0026 | 0.9122 ± 0.0030 | OOM | OOM | OOM |
| | **LGNSDE (Ours)** | 0.9850 ± 0.0010 | 0.9426 ± 0.0057 | 0.9892 ± 0.0001 | 0.9942 ± 0.0002 | 0.9338 ± 0.0050 |
| **AURC (↓)** | GCN | 0.0391 ± 0.0037 | 0.1104 ± 0.0065 | 0.0425 ± 0.0024 | 0.0153 ± 0.0010 | 0.1051 ± 0.0215 |
| | GNODE | 0.0428 ± 0.0050 | 0.1136 ± 0.0056 | 0.0658 ± 0.0080 | 0.0206 ± 0.0022 | 0.1147 ± 0.0199 |
| | BGCN | 0.0383 ± 0.0045 | 0.2849 ± 0.0595 | 0.0733 ± 0.0176 | 0.0200 ± 0.0053 | 0.3716 ± 0.0681 |
| | ENSEMBLE | 0.0524 ± 0.0080 | 0.1402 ± 0.0081 | 0.0353 ± 0.0016 | 0.0188 ± 0.0010 | 0.1053 ± 0.0089 |
| | GRAPH GP | 0.1007 ± 0.0039 | 0.2022 ± 0.0043 | OOM | OOM | OOM |
| | **LGNSDE (Ours)** | 0.0353 ± 0.0027 | 0.1199 ± 0.0110 | 0.0378 ± 0.0010 | 0.0139 ± 0.0007 | 0.0980 ± 0.0094 |
| **Accuracy (↑)** | GCN | 0.8806 ± 0.0065 | 0.7877 ± 0.0077 | 0.8850 ± 0.0045 | 0.9355 ± 0.0020 | 0.7995 ± 0.0238 |
| | GNODE | 0.8710 ± 0.0091 | 0.7836 ± 0.0086 | 0.8708 ± 0.0048 | 0.9302 ± 0.0047 | 0.7799 ± 0.0280 |
| | BGCN | 0.8728 ± 0.0089 | 0.5167 ± 0.0727 | 0.8208 ± 0.0246 | 0.9104 ± 0.0199 | 0.4788 ± 0.0707 |
| | ENSEMBLE | 0.8627 ± 0.0063 | 0.7530 ± 0.0075 | 0.8996 ± 0.0020 | 0.9410 ± 0.0010 | 0.7924 ± 0.0119 |
| | GRAPH GP | 0.7868 ± 0.0061 | 0.7129 ± 0.0051 | OOM | OOM | OOM |
| | **LGNSDE (Ours)** | 0.8889 ± 0.0067 | 0.7826 ± 0.0099 | 0.8984 ± 0.0021 | 0.9381 ± 0.0017 | 0.8208 ± 0.0111 |

Table 5: Results of the active learning experiments **using a maximum entropy acquisition function**. The table reports AUROC, AURC, and Accuracy (Mean ± Std) for all datasets. Red indicates the best-performing model, and blue indicates the second-best-performing model.

Pubmed from 0.8097 to 0.8208, compared to the random acquisition setup. This shows that using uncertainty to guide label selection leads to better overall performance.

In summary, the total entropy acquisition function helps LGNSDE perform more effectively, particularly by selecting more informative data points than random selection, resulting in higher accuracy and better uncertainty management. For details on the effects of the hyperparameters and additional benchmark experiments, please refer to Appendix C.

# 6 RELATED WORK

Uncertainty quantification in GNNs has recently gained attention, with contributions from Bayesian GNNs (Hasanzadeh et al., 2020), ensemble-based methods (Lin et al., 2022), and Gaussian Processes on graphs (Borovitskiy et al., 2021; Sáez de Ocáriz Borde et al., 2024). We benchmark our LGNSDE framework against these methods and show improved performance across tasks, demonstrating more flexibility in capturing both aleatoric and epistemic uncertainty through a dynamic stochastic framework.

The work of Bishnoi et al. (2023) introduces stochastic elements into graph learning but is restricted to learning scalar parameters for an SDE, unlike our method, which models the graph evolution directly as an SDE. Similarly, Poli et al. (2021) attempt to propose Graph Neural SDEs with a different formulation that uses a finite-dimensional KL divergence in the ELBO instead of an infinite-dimensional version. They also use prior distributions instead of prior processes, which are better suited for modelling continuous dynamical systems. Hence, constructing a different method. Moreover, they do not provide theoretical analysis or thoroughly explore the model's uncertainty estimation capabilities beyond limited toy experiments.

The framework proposed by Lin et al. (2024) employs stochastic partial differential equations (SPDEs) to model message passing for uncertainty estimation, leveraging a novel Q-Wiener process to propagate uncertainty directly within the graph diffusion process. While their approach emphasizes diffusion-based uncertainty, our method adopts a Bayesian framework with SDEs, focusing on quantifying uncertainty in the latent space instead. Furthermore, the work by Stadler et al. (2021) on Graph Posterior Networks (GPN) takes a Bayesian approach to uncertainty estimation, focusing on interdependent nodes in graph-structured data. Their model explicitly performs posterior updates for node-level classification by leveraging Dirichlet distributions and performs well in uncertainty-sensitive tasks. However, GPNs do not capture the dynamic evolution of node embeddings over time as our LGNSDE does. Lastly, (Zhao et al., 2020) work on uncertainty-aware semi-supervised learning for graphs introduces a method which uses belief theory to quantify uncertainty types like vacuity and dissonance. Their focus on semi-supervised learning and belief theory-based uncertainty

contrasts with our SDE-based approach, where uncertainty is captured through variance in latent representations and modelled using Bayesian updates.

## 7 CONCLUSIONS AND FUTURE WORK

We have introduced Latent Graph Neural Stochastic Differential Equations (LGNSDE), a novel framework designed to quantify uncertainty in graph-structured data. By leveraging both epistemic and aleatoric uncertainty through a Bayesian prior-posterior mechanism and Brownian motion, LGNSDE provides meaningful uncertainty estimates. Theoretical guarantees demonstrate that our model ensures well-posedness, variance bounds, and robustness to small perturbations in inputs. Empirically, LGNSDE performs competitively across a range of benchmarks, excelling in out-of-distribution detection, noise robustness, and active learning tasks. Future directions include exploring higher-order SDEs, optimizing computational efficiency, and applying the model to real-world systems requiring reliable uncertainty quantification such as recommendation systems, drug discovery or dynamic networks.

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

# A  THEORETICAL REMARKS

## A.1  NOTATION

Let $\mathcal{G} = (\mathcal{V}, \mathcal{E})$ denote a graph with node set $\mathcal{V}$ and edge set $\mathcal{E}$. The node feature matrix at time $t$ is $\mathbf{H}(t) \in \mathbb{R}^{n \times d}$, where $n$ is the number of nodes and $d$ is the feature dimension. The evolution of $\mathbf{H}(t)$ is described by a Graph Neural Stochastic Differential Equation, with drift function $\mathbf{F}_{\mathcal{G}}(\mathbf{H}(t), t, \boldsymbol{\theta})$ and diffusion function $\mathbf{G}_{\mathcal{G}}(\mathbf{H}(t), t)$. Here, $\mathbf{F}_{\mathcal{G}}$ depends on the graph $\mathcal{G}$, the node features $\mathbf{H}(t)$, time $t$, and parameters $\boldsymbol{\theta}$. The diffusion function $\mathbf{G}_{\mathcal{G}}$ depends on $\mathcal{G}$ and $\mathbf{H}(t)$ but not on $\boldsymbol{\theta}$, as in practice, this is usually a constant function. The randomness is introduced through the Brownian motion $\mathbf{W}(t)$.

The constants $L_f$ and $L_g$ are Lipschitz constants for the drift and diffusion functions, respectively, ensuring the existence and uniqueness of the solution to the GNSDE. The linear growth condition is controlled by a constant $K$, preventing unbounded growth in $\mathbf{F}_{\mathcal{G}}$ and $\mathbf{G}_{\mathcal{G}}$. Finally, $Var(\mathbf{H}(t))$ represents the variance of the node features, capturing the aleatoric uncertainty in the system, which is also reflected in the variance of the model output $\hat{\mathbf{y}}(t) = \mathbf{h}(\mathbf{H}(t))$.

## A.2  TECHNICAL ASSUMPTIONS

**Assumption 1.** *The drift and diffusion functions $\mathbf{F}_{\mathcal{G}}$ and $\mathbf{G}_{\mathcal{G}}$ satisfy the following Lipschitz conditions:*

$$\|\mathbf{F}_{\mathcal{G}}(\mathbf{H}_1(t), t, \boldsymbol{\theta}) - \mathbf{F}_{\mathcal{G}}(\mathbf{H}_2(t), t, \boldsymbol{\theta})\|_F \leq L_f \|\mathbf{H}_1(t) - \mathbf{H}_2(t)\|_F \tag{3}$$

$$\|\mathbf{G}_{\mathcal{G}}(\mathbf{H}_1(t), t) - \mathbf{G}_{\mathcal{G}}(\mathbf{H}_2(t), t)\|_F \leq L_g \|\mathbf{H}_1(t) - \mathbf{H}_2(t)\|_F \tag{4}$$

*for all $\mathbf{H}_1, \mathbf{H}_2 \in \mathbb{R}^{n \times d}$, $t \in [0, T]$, and some constants $L_f$ and $L_g$.*

**Assumption 2.** *The drift and diffusion functions $\mathbf{F}_{\mathcal{G}}$ and $\mathbf{G}_{\mathcal{G}}$ satisfy a linear growth condition:*

$$\|\mathbf{F}_{\mathcal{G}}(\mathbf{H}(t), t, \boldsymbol{\theta})\|_F^2 + \|\mathbf{G}_{\mathcal{G}}(\mathbf{H}(t), t)\|_F^2 \leq K(1 + \|\mathbf{H}(t)\|_F^2),$$

*for all $\mathbf{H} \in \mathbb{R}^{n \times d}$, $t \in [0, T]$, and some constant $K$.*

**Assumption 3.** *The variance of the initial conditions, $\mathbf{H}(0) = \mathbf{H}_0$, of the dynamical system is bounded: $\mathbb{E}[\|\mathbf{H}_0\|_F^2] < \infty$.*

## A.3  PROOFS

**Proposition 1.** *Under assumptions 1-3 and given Theorem 1, there exists a unique mild[4] solution to an LGNSDE of the form*

$$d\mathbf{H}(t) = \mathbf{F}_{\mathcal{G}}(\mathbf{H}(t), t, \boldsymbol{\theta}) \, dt + \mathbf{G}_{\mathcal{G}}(\mathbf{H}(t), t) \, d\mathbf{W}(t),$$

*whose variance bounds the variance of the model output $\hat{\mathbf{y}}(t)$ as:*

$$Var(\hat{\mathbf{y}}(t)) \leq L_h^2 Var(\mathbf{H}(t)),$$

*where $L_h^2$ is the Lipschitz constant of the readout layer. This ensures that the output variance is bounded by the prior variance of the latent space, providing a controlled measure of uncertainty.*

*Proof.* Using the result in 1 (Lin et al. (2024)), it follows that the Lipschitz conditions of $\mathbf{F}_{\mathcal{G}}$ and $\mathbf{G}_{\mathcal{G}}$ ensure the existence and uniqueness of a mild solution $\mathbf{H}(t)$ to the GNSDE.

Now, consider the stochastic part of the variance of the solution. By applying the Itô isometry, we can compute the expectation of the Frobenius norm of the stochastic integral:

$$\mathbb{E}\left[\left\|\int_0^t \mathbf{G}_{\mathcal{G}}(\mathbf{H}(u), u) d\mathbf{W}(u)\right\|_F^2\right] = \mathbb{E}\left[\int_0^t \|\mathbf{G}_{\mathcal{G}}(\mathbf{H}(u), u)\|_F^2 du\right].$$

---

[4] A mild solution to an SDE is expressed via an integral equation involving the semigroup generated by the linear operator and represents a weaker notion of the solution.

Under the Lipschitz condition on $\mathbf{G}_\mathcal{G}$, we can bound the variance of $\mathbf{H}(t)$ as follows:

$$\text{Var}(\mathbf{H}(t)) = \int_0^t \|\mathbf{G}_\mathcal{G}(\mathbf{H}(u), u)\|_F^2 \, du.$$

If $\mathbf{G}_\mathcal{G}$ is bounded, i.e., $\|\mathbf{G}_\mathcal{G}(\mathbf{H}(u), u)\|_F \leq M$ for some constant $M$, then $\text{Var}(\mathbf{H}(t)) \leq M^2 t$. This shows that the variance of the latent state $\mathbf{H}(t)$ is bounded and grows linearly with time, capturing the aleatoric uncertainty introduced by the stochastic process.

Finally, assuming that the model output $\hat{\mathbf{y}}(t)$ is a function of the latent state $\mathbf{H}(t)$, $\hat{\mathbf{y}}(t) = \mathbf{h}(\mathbf{H}(t))$, where $\mathbf{h} : \mathbb{R}^{n \times d} \to \mathbb{R}^{n \times p}$ is a smooth function, we can apply Itô's Lemma as follows:

$$dy(t) = h'(\mathbf{H}(t)) \left[ \mathbf{F}_\mathcal{G}(\mathbf{H}(t), t, \boldsymbol{\theta}) \, dt + \mathbf{G}_\mathcal{G}(\mathbf{H}(t), t) \, d\mathbf{W}(t) \right] + \frac{1}{2} h''(\mathbf{H}(t)) \mathbf{G}_\mathcal{G}(\mathbf{H}(t), t)^2 \, dt.$$

For the variance of $\hat{\mathbf{y}}(t)$, we focus on the term involving $\mathbf{G}_\mathcal{G}(\mathbf{H}(t), t) \, d\mathbf{W}(t)$:

$$\text{Var}(\hat{\mathbf{y}}(t)) = \int_0^t \text{tr}\left( \mathbf{h}'(\mathbf{H}(u))^\top \mathbf{G}_\mathcal{G}(\mathbf{H}(u), u) \mathbf{G}_\mathcal{G}(\mathbf{H}(u), u)^\top \mathbf{h}'(\mathbf{H}(u)) \right) \, du.$$

Using the Cauchy-Schwarz inequality for matrix norms, we can bound this as follows:

$$\text{tr}\left( \mathbf{h}'(\mathbf{H}(u))^\top \mathbf{G}_\mathcal{G}(\mathbf{H}(u), u) \mathbf{G}_\mathcal{G}(\mathbf{H}(u), u)^\top \mathbf{h}'(\mathbf{H}(u)) \right) \leq \|\mathbf{h}'(\mathbf{H}(u))\|_F^2 \|\mathbf{G}_\mathcal{G}(\mathbf{H}(u), u)\|_F^2.$$

Therefore, if $\mathbf{h}$ is Lipschitz continuous with constant $L_h$, then:

$$\text{Var}(\mathbf{y}(t)) \leq L_h^2 \int_0^t \|\mathbf{G}_\mathcal{G}(\mathbf{H}(u), u)\|_F^2 \, du = L_h^2 \text{Var}(\mathbf{H}(t)).$$

Hence, under the Lipschitz continuity and boundedness assumptions for the drift and diffusion functions, the solution to the GNSDE exists and is unique, and its output variance serves as a meaningful measure of aleatoric uncertainty. $\square$

**Proposition 2.** *Under assumptions 1-3, consider two initial conditions $\mathbf{H}_0$ and $\tilde{\mathbf{H}}_0 = \mathbf{H}_0 + \delta\mathbf{H}(0)$, where $\delta\mathbf{H}(0) \in \mathbb{R}^{n \times d}$ is a small perturbation in the initial node features with $\|\delta\mathbf{H}(0)\|_F = \epsilon$. Assume that $\mathbf{H}_0$ is taken from a compact set $\mathcal{H} \subseteq \mathbb{R}^{n \times d}$. Then, the deviation between the solutions $\mathbf{H}(t)$ and $\tilde{\mathbf{H}}(t)$ of the LGNSDE with these initial conditions remains bounded across time $t$[5], specifically*

$$\mathbb{E}[\|\mathbf{H}(t) - \tilde{\mathbf{H}}(t)\|_F] \leq \epsilon e^{(L_f + \frac{1}{2} L_g^2)t}.$$

*Proof.* Consider two solutions $\mathbf{H}_1(t)$ and $\mathbf{H}_2(t)$ of the GNSDE with different initial conditions. Define the initial perturbation as $\delta\mathbf{H}(0)$ where $\mathbf{H}_1(0) = \mathbf{H}_0 + \delta\mathbf{H}(0)$ and $\mathbf{H}_2(0) = \mathbf{H}_0$, with $\|\delta\mathbf{H}(0)\|_F = \epsilon$.

The difference between the two solutions at any time $t$ is given by $\delta\mathbf{H}(t) = \mathbf{H}_1(t) - \mathbf{H}_2(t)$. The dynamics of $\delta\mathbf{H}(t)$ are:

$$d(\delta\mathbf{H}(t)) = \left[ \mathbf{F}_\mathcal{G}(\mathbf{H}_1(t), t, \boldsymbol{\theta}) - \mathbf{F}_\mathcal{G}(\mathbf{H}_2(t), t, \boldsymbol{\theta}) \right] dt + \left[ \mathbf{G}_\mathcal{G}(\mathbf{H}_1(t), t) - \mathbf{G}_\mathcal{G}(\mathbf{H}_2(t), t) \right] d\mathbf{W}(t).$$

Applying Itô's lemma to $\text{tr}(\delta\mathbf{H}(t)^\top \delta\mathbf{H}(t))$, we obtain:

$$
\begin{aligned}
d(\text{tr}(\delta\mathbf{H}(t)^\top \delta\mathbf{H}(t))) = {} & 2\text{tr}\left( \delta\mathbf{H}(t)^\top \left[ \mathbf{F}_\mathcal{G}(\mathbf{H}_1(t), t, \boldsymbol{\theta}) - \mathbf{F}_\mathcal{G}(\mathbf{H}_2(t), t, \boldsymbol{\theta}) \right] \right) dt \\
& + 2\text{tr}\left( \delta\mathbf{H}(t)^\top \left[ \mathbf{G}_\mathcal{G}(\mathbf{H}_1(t), t) - \mathbf{G}_\mathcal{G}(\mathbf{H}_2(t), t) \right] d\mathbf{W}(t) \right) \\
& + \text{tr}\left( \left[ \mathbf{G}_\mathcal{G}(\mathbf{H}_1(t), t) - \mathbf{G}_\mathcal{G}(\mathbf{H}_2(t), t) \right]^\top \left[ \mathbf{G}_\mathcal{G}(\mathbf{H}_1(t), t) - \mathbf{G}_\mathcal{G}(\mathbf{H}_2(t), t) \right] \right) dt.
\end{aligned}
$$

---

[5]Note that while the bound is exponential in $t$, in practice, the time horizon is usually constrained to a limited range, such as $t \in [0, 1]$. Within this interval, the exponential factor does not grow excessively, ensuring that the deviation between the perturbed and unperturbed solutions remains under control.

Taking the expected value, the stochastic integral term involving $d\mathbf{W}(t)$ has an expectation of zero due to the properties of the Brownian motion. Thus, we have:

$$\mathbb{E}[d(\text{tr}(\delta\mathbf{H}(t)^\top\delta\mathbf{H}(t)))] = \mathbb{E}\left[2\text{tr}(\delta\mathbf{H}(t)^\top[\mathbf{F}_\mathcal{G}(\mathbf{H}_1(t),t,\boldsymbol{\theta}) - \mathbf{F}_\mathcal{G}(\mathbf{H}_2(t),t,\boldsymbol{\theta})])\right]\,dt$$
$$+ \mathbb{E}[\|\mathbf{G}_\mathcal{G}(\mathbf{H}_1(t),t) - \mathbf{G}_\mathcal{G}(\mathbf{H}_2(t),t)\|_F^2]\,dt.$$

Here, the second term arises from the variance of the diffusion term, as captured by Itô's Lemma. Using the Lipschitz bounds for $\mathbf{F}_\mathcal{G}$ and $\mathbf{G}_\mathcal{G}$, we obtain:

$$\mathbb{E}[d(\text{tr}(\delta\mathbf{H}(t)^\top\delta\mathbf{H}(t)))] \leq \left(2L_f\mathbb{E}[\text{tr}(\delta\mathbf{H}(t)^\top\delta\mathbf{H}(t))] + L_g^2\mathbb{E}[\text{tr}(\delta\mathbf{H}(t)^\top\delta\mathbf{H}(t))]\right)dt.$$

Rewriting this as a differential inequality:

$$\frac{d}{dt}\mathbb{E}[\text{tr}(\delta\mathbf{H}(t)^\top\delta\mathbf{H}(t))] \leq (2L_f + L_g^2)\mathbb{E}[\text{tr}(\delta\mathbf{H}(t)^\top\delta\mathbf{H}(t))].$$

Solving this using Gronwall's inequality gives:

$$\mathbb{E}[\text{tr}(\delta\mathbf{H}(t)^\top\delta\mathbf{H}(t))] \leq \text{tr}(\delta\mathbf{H}(0)^\top\delta\mathbf{H}(0))e^{(2L_f+L_g^2)t}.$$

Since $\|\delta\mathbf{H}(0)\|_F = \epsilon$, we conclude that:

$$\mathbb{E}[\|\delta\mathbf{H}(t)\|_F] \leq \epsilon e^{(L_f+\frac{1}{2}L_g^2)t}.\text{[6]}$$

Hence, the deviation in the output remains bounded under small perturbations to the initial conditions, providing robustness guarantees. $\square$

### A.4 LGNSDE AS A CONTINUOUS REPRESENTATION OF GRAPH RESNET WITH STOCHASTIC NOISE INSERTION

Consider a Latent Graph Neural Stochastic Differential Equation (LGNSDE) represented as

$$d\mathbf{H}(t) = \mathbf{F}_\mathcal{G}(\mathbf{H}(t),t)dt + \mathbf{G}_\mathcal{G}(\mathbf{H}(t),t)d\mathbf{W}(t),$$

where $\mathbf{H}(t) \in \mathbb{R}^{n\times d}$, $\mathbf{F}_\mathcal{G}(\mathbf{H}(t),t)$, and $\mathbf{G}_\mathcal{G}(\mathbf{H}(t),t)$ are matrix-valued functions, and $\mathbf{W}(t)$ is a Brownian motion. The numerical Euler-Maruyama discretization of this GNSDE can be expressed as

$$\frac{\mathbf{H}(t_{j+1}) - \mathbf{H}(t_j)}{\Delta t} \approx \mathbf{F}_\mathcal{G}(\mathbf{H}(t_j),t_j) + \frac{\mathbf{G}_\mathcal{G}(\mathbf{H}(t_j),t_j)\Delta\mathbf{W}_j}{\Delta t},$$

which simplifies to

$$\mathbf{H}_{j+1} = \mathbf{H}_j + \mathbf{F}_\mathcal{G}(\mathbf{H}_j,t_j)\Delta t + \mathbf{G}_\mathcal{G}(\mathbf{H}_j,t_j)\Delta\mathbf{W}_j.$$

Here, $\Delta t$ represents a fixed time step and $\Delta\mathbf{W}_j$ is a Brownian increment, normally distributed with mean zero and variance $\Delta t$. This numerical discretization is analogous to a Graph Recurrent Network (Graph ReNet) with a specific structure, where Brownian noise is injected at each recurrent layer. Therefore, the Graph Neural SDE can be interpreted as a deep Graph ReNet where the depth corresponds to the number of discretization steps of the SDE solver.

---

[6]Note that the second term (stochastic part) can be omitted as the first term dominates.

## B DETAILS OF THE EXPERIMENTAL SETUP

### B.1 HYPERPARAMETER SEARCH

Table 6: Hyperparameter Grid Search Configuration

| Hyperparameter | Values |
|---|---|
| Learning Rate | {0.001, 0.005, 0.01, 0.1} |
| Weight Decay | {0.01, 0.001, 0.0005, 0.0001} |
| Epoch | {15, 100, 200, 300} |
| Dropout | {0.0, 0.1, 0.3, 0.5} |
| Hidden Dimension | {16, 32, 64, 128, 256} |
| Step Size | {0.01, 0.05, 0.1, 0.2} |

Table 7: Hyperparameters left out of the grid search for all models and used for all datasets.

| Parameter | GNSDE | GNODE | Other |
|---|---|---|---|
| $t_1$ | 1 | 1 | N/A |
| Optimizer | Adam | Adam | Adam |
| Method | SRK | RK4 | N/A |
| Early Stop | 20 | 20 | 20 |
| Diffusion $\mathbf{G}_{\mathcal{G}}$ | 1.0 | N/A | N/A |

### B.2 ACTIVE LEARNING

Table 8: Dataset statistics before and after active learning.

| Dataset | # Nodes | # Links | Training/Validation/Test Split | Initial # Training Labels | Final # Training Labels |
|---|---|---|---|---|---|
| Cora | 2,708 | 5,429 | 140/500/1000 | 140 | 280 |
| Citeseer | 3,327 | 4,732 | 120/500/1000 | 120 | 240 |
| Computers | 13,752 | 245,861 | 200/500/1000 | 200 | 400 |
| Photo | 7,650 | 119,081 | 160/500/1000 | 160 | 320 |
| Pubmed | 19,717 | 44,338 | 60/500/1000 | 60 | 120 |

### B.3 LGNSDE HYPERPARAMETERS

Table 9: Accuracy vs. prior drift in Cora.

| Prior drift | Accuracy (%) |
|---|---|
| -100.0 | 81.29 |
| -10.0 | 82.21 |
| -5.0 | 80.52 |
| -1.0 | 82.17 |
| -0.5 | 83.66 |
| 0.0 | 82.74 |
| 0.5 | 82.58 |
| 1.0 | 82.41 |
| 5.0 | 80.48 |
| 10.0 | 80.80 |
| 20.0 | 82.74 |
| 100.0 | 82.17 |

Table 10: Accuracy vs. prior diffusion in Cora.

| Sigma ($\sigma$) | Accuracy (%) |
|---|---|
| 0.1 | 79.45 |
| 0.2 | 80.13 |
| 0.5 | 81.32 |
| 0.8 | 80.52 |
| 1.0 | 80.79 |
| 1.5 | 78.93 |
| 2.0 | 78.47 |
| 5.0 | 76.85 |
| 7.0 | 68.78 |
| 10.0 | 60.54 |

## C  MORE EXPERIMENTS AND BENCHMARKS

### C.1  LGNSDE BACKBONE MODEL GCN VS GAT

| Metric | Model | Cora | Citeseer |
|--------|-------|------|----------|
| **AUROC (↑)** | **LGNSDE-GAT** | 0.8249 ± 0.0321 | 0.8003 ± 0.0488 |
| | LGNSDE-GCN | 0.7614 ± 0.0804 | 0.8258 ± 0.0418 |
| **AURC (↓)** | **LGNSDE-GAT** | 0.0108 ± 0.0011 | 0.0536 ± 0.0080 |
| | LGNSDE-GCN | 0.0168 ± 0.0070 | 0.0479 ± 0.0109 |
| **Accuracy (↑)** | **LGNSDE-GAT** | 0.9474 ± 0.0002 | 0.8642 ± 0.0090 |
| | LGNSDE-GCN | 0.9471 ± 0.0003 | 0.8729 ± 0.0108 |

Table 11: AUROC (Mean ± Std) and AURC (Mean ± Std) for OOD Detection across datasets. Red denotes the best-performing model, and blue denotes the second-best-performing model.

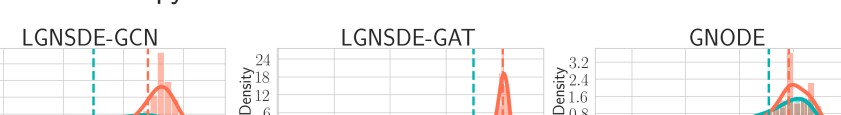

Figure 4: Illustrating two Drift function backbone the standard GCN vs Graph Attention Network.

### C.2  COMPUTATIONAL COST COMPARISON

| Model | Time per Epoch (s) |
|-------|--------------------|
| GCN | 0.19 |
| GAP | 0.23 |
| BGCN | 0.36 |
| GNODE | 0.38 |
| GRAPH GP | 0.72 |
| GNSD | 0.92 |
| ENSEMBLE | 1.13 |
| **LGNSDE (Ours)** | **1.23** |

Table 12: Time Taken per Epoch for Different Models .

### C.3  FURTHER BENCHMARKING AND COMPLEXITIES

Table 13: OOD detection performance comparison on Cora with different OOD constructions.

| Model | Label leave-out | | | | | Feature perturbation | | | | |
|-------|-------|---------|----------|-------|---------|-------|---------|----------|-------|---------|
| | AUROC | AUPR in | AUPR out | FPR95 | DET ACC | AUROC | AUPR in | AUPR out | FPR95 | DET ACC |
| GCN | 88.67 | 74.52 | 95.87 | 52.40 | 78.80 | 75.40 | 54.22 | 86.69 | 80.98 | 69.94 |
| GAT | 90.81 | 71.97 | 96.50 | 44.91 | 83.65 | 89.39 | 80.19 | 92.80 | 70.37 | 76.16 |
| GRAND | 88.58 | 75.72 | 94.28 | 63.59 | 82.34 | 87.04 | 74.81 | 93.42 | 55.35 | 76.43 |
| GREAD | 88.40 | 73.80 | 95.01 | 57.35 | 80.74 | 86.49 | 70.59 | 93.01 | 57.10 | 75.30 |
| MSP | 91.58 | 80.59 | 96.72 | 44.03 | 84.51 | 90.58 | 82.31 | 92.02 | 50.33 | 80.45 |
| ODIN | 49.24 | 24.27 | 75.45 | 100.00 | 49.95 | 49.80 | 26.92 | 72.95 | 100.00 | 49.98 |
| Mahalanobis | 66.83 | 36.69 | 85.92 | 82.96 | 59.62 | 60.46 | 40.65 | 74.52 | 99.59 | 58.52 |
| GNNsafe | 92.68 | 82.03 | 97.48 | 31.54 | 83.48 | 93.28 | 88.16 | 96.35 | 43.43 | 83.71 |
| GCN-Ensemble | 90.97 | 80.35 | 97.37 | 29.92 | 85.73 | 89.01 | 79.71 | 94.57 | 61.74 | 74.57 |
| BGCN | 91.16 | 79.41 | 96.60 | 46.30 | 84.40 | 84.82 | 75.40 | 91.20 | 75.20 | 78.09 |
| GKDE | 82.80 | 65.12 | 92.62 | 70.59 | 71.64 | 80.22 | 63.05 | 90.01 | 75.01 | 64.74 |
| GPN | 90.10 | 79.98 | 96.13 | 50.71 | 82.81 | 91.89 | 82.62 | 96.01 | 42.32 | 81.56 |
| GNSD | 94.76 | 88.45 | 97.73 | 27.38 | 89.74 | 91.77 | 94.41 | 88.23 | 26.27 | 86.92 |
| **LGNSDE (Ours)** | 93.14 | 85.01 | 96.65 | 49.03 | 81.42 | 90.61 | 86.35 | 92.77 | 31.63 | 83.05 |

Table 14: OOD detection comparison on Amazon-Computers with different OOD constructions.

| Model | Label leave-out | | | | | Feature perturbation | | | | |
|---|---|---|---|---|---|---|---|---|---|---|
| | AUROC | AUPR in | AUPR out | FPR95 | DET ACC | AUROC | AUPR in | AUPR out | FPR95 | DET ACC |
| GCN | 82.35 | 56.46 | 93.67 | 56.06 | 74.72 | 80.55 | 78.53 | 78.55 | 80.67 | 75.09 |
| GAT | 80.66 | 53.19 | 93.05 | 53.91 | 72.65 | 73.69 | 78.00 | 65.61 | 97.41 | 75.76 |
| GRAND | 80.27 | 52.51 | 92.84 | 54.81 | 71.99 | 84.93 | 81.29 | 87.33 | 54.98 | 65.34 |
| GREAD | 80.56 | 54.05 | 92.70 | 54.14 | 72.68 | 85.38 | 79.07 | 87.60 | 59.10 | 68.29 |
| MSP | 74.88 | 47.53 | 89.64 | 75.52 | 68.85 | 72.86 | 74.50 | 67.73 | 95.70 | 70.81 |
| ODIN | 71.78 | 37.70 | 89.87 | 70.54 | 50.18 | 79.13 | 80.09 | 77.09 | 83.09 | 66.75 |
| Mahalanobis | 71.87 | 37.76 | 89.87 | 70.24 | 50.18 | 74.47 | 67.54 | 76.28 | 82.48 | 50.04 |
| GNNsafe | 90.50 | 77.20 | 95.05 | 48.25 | 84.47 | 89.46 | 95.17 | 84.01 | 75.62 | 76.49 |
| GCN-Ensemble | 79.53 | 52.39 | 91.99 | 69.28 | 73.51 | 77.71 | 79.45 | 72.60 | 94.20 | 77.51 |
| BGCN | 82.19 | 57.52 | 93.30 | 57.43 | 73.55 | 83.60 | 82.93 | 81.50 | 72.49 | 75.78 |
| GKDE | 76.46 | 48.18 | 90.64 | 73.36 | 64.35 | 71.69 | 71.40 | 69.04 | 90.83 | 69.70 |
| GPN | 88.76 | 68.23 | 96.45 | 42.08 | 81.02 | 87.92 | 85.99 | 85.98 | 67.10 | 81.24 |
| GNSD | 94.06 | 82.27 | 97.06 | 31.47 | 88.76 | 95.95 | 94.76 | 94.62 | 15.69 | 91.28 |
| **LGNSDE (Ours)** | 90.71 | 74.25 | 97.78 | 68.20 | 82.28 | 90.86 | 93.64 | 84.00 | 35.21 | 82.49 |

Table 15: Time and Memory Complexity of Models

| Model | Time Complexity | Memory Complexity |
|---|---|---|
| Graph Posterior Network (GPN) | $O(N \cdot K)$ | $O(N \cdot C)$ |
| Graph Gaussian Processes (GGP) | $O(NM^2)$ | $O(NM)$ |
| Graph Neural ODEs (GNODE) | $O(E \cdot F^2 \cdot \text{NFEs})$ | $O(1)$ |
| Latent Graph Neural SDEs (LGNSDE) | $O(L \log L(|E|d + |V|d))$ | $O(1)$ |

Table 16: Explanation of Time and Memory Complexity Components

| Component | Description |
|---|---|
| $N$ | Number of nodes in the graph |
| $K$ | Average degree of the nodes (used in GPN) |
| $M$ | Number of inducing points (used in GGP to reduce complexity) |
| $E$ | Number of edges in the graph |
| $F$ | Feature dimensionality for nodes or edges |
| NFEs | Number of function evaluations during ODE solving (for GNODE) |
| $L$ | Number of steps required by the SDE or ODE solver (for LGNSDE) |
| $C$ | Number of classes in classification tasks (for GPN) |

