# OpenReview forum: "Uncertainty Modeling in Graph Neural Networks via Stochastic Differential Equations"
_ICLR.cc/2025/Conference — ICLR 2025 Spotlight_

### Official Review · Reviewer_9vdg · 2024-10-23

**Soundness:** 2
**Presentation:** 3
**Contribution:** 2
**Rating:** 6
**Confidence:** 3

**Summary:**

This paper introduces a new model (LGNSDE) for learning on graphs with uncertainty quantification. It leverages an Ornstein-Uhlenbeck prior and a posterior with drift parameterized by a GCN model that is trained through variational inference. This SDE approach is latent as it operates on node feature embeddings. Two novel theoretical results are shown, providing a bound on model output variance and a bound on solutions under node feature perturbations. Six graph models are then compared in various experiments on five standard graph datasets, including studies of OOD detection, test noise, and active learning. The proposed model performs favorably across these experiments.

**Strengths:**

- The LGNSDE model description is clear, concise, and intuitive.
- The two theoretical results are important and original, providing a good deal of strength to the proposed methodology.
- The experiments clearly show this model has significant potential in delivering upon the promise of uncertainty quantification for graph-structured learning problems.

**Weaknesses:**

- Some of the cited works (particularly Calvo-Ordoñez et al., 2024, and Xu et al., 2022) leverage SDEs in a very similar way, but do not consider the problem of learning on graphs. While the application to graphs is creative, the model definition is somewhat limited in novelty.
- The choice of a constant drift and diffusion function in the OU prior is not sufficiently explored. It would be great if there was mention of why this is a reasonable restriction if this is indeed the case.
- The experimental section would benefit from increased clarity and specificity, especially in regard to the structure and setup of experiments. Choices such as number of epochs and early stopping criteria, if any, are not explicitly stated.
- It is stated in Section 5 that hyperparameters that achieved the highest validation accuracy were chosen. It is unclear to me which validation accuracy is used here, and knowing the exact grid search setup would be ideal for improved reproducibility.
- Table 7 shows that uniform hyperparameters were chosen across all models; if the hyperparameter choice was performed via search for each model independently, the comparison results might be more fair.
- The most impactful perceived weakness of this paper was the lack of experimental comparison with the models referenced as related works, particularly those referenced in Section 6. The other methods used in comparison do not follow a stochastic approach, limiting the ability for a fair comparison. I would be extremely interested to see a comparative study against the referenced GNSD (Lin et al., 2024) method, for example.
- For OOD detection, comparison against methods built specifically for this task on graphs, like GNNSafe (Wu et al., 2023) would be enlightening.
- One of the theoretical claims was a resistance to small perturbations in graph structure. This might lead to a compelling experiment, but this claim is not tested.

There are two small writing nitpicks:
- In the first line of the introduction, "Before the widespread of neural networks" is missing a word. Perhaps use "widespread success" or similar?
- In Section 6, "Hence, constructing a different method." does not read as a complete sentence, and should probably be folded into the line before it.

**Questions:**

- Is it possible to allow a non-constant drift function in the prior OU process? Would a similar method or extension allow this?
- What is the effect upon training and convergence for different choices of hyperparameters, and in particular, for different choices of constant drift and diffusion functions? Has an ablation study been performed for this proposed model?
- How does the walltime of learning with an LGNSDE compare to walltime for the other tested methods?
- Are there any interesting limitations on the practicality of this method, and in your experience, what can allow for learning the best results?

---

> ### Author Response · Authors · 2024-11-20
> **Official Response to Reviewer 9vdg (Part 1)**
>
> Dear Reviewer 9vdg, thank you for your thoughtful and detailed review. We appreciate your recognition of the clarity and strength of our model description, the originality of our theoretical results, and the potential impact of our work on uncertainty quantification for graph-structured learning. We hope to have carefully addressed your concerns and made significant updates to the manuscript based on your feedback. We believe that our responses and the revised submission should address your concerns and demonstrate the strengths and contributions of our work. If not, please let us know so that we can address any remaining concerns.
>
> Find below more details on our responses.
>
> ---
>
> > Some of the cited works (particularly Calvo-Ordoñez et al., 2024, and Xu et al., 2022) leverage SDEs in a very similar way...
>
> We appreciate the reviewer's comment and the chance to clarify our unique contributions. While Calvo-Ordoñez et al. (2024) and Xu et al. (2022) use SDEs to model weight trajectories in Bayesian Neural Networks, our work focuses on modeling uncertainty in the latent space of Graph Neural Networks. Our approach is fundamentally different in that it addresses the specific problem of learning on graph-structured data.
>
> Quantifying the uncertainty in the latent space of GNNs allows us to provide theoretical guarantees and motivation on well-posedness, robustness, and the connection between latent space variance and output uncertainty. Which, to us, is not translatable to a framework where one uses SDEs in weight space. Please, also note that these contributions are further supported by strong empirical results in tasks like out-of-distribution detection, noise perturbation and active learning.
>
> ---
>
> > The choice of a constant drift and diffusion function in the OU prior is not sufficiently explored...
>
> We thank the reviewer for pointing this out. The choice of a constant drift and diffusion function in the Ornstein-Uhlenbeck (OU) prior is analogous to using a standard Gaussian prior in Bayesian models (e.g. BNNs, GPs, etc) to reflect the lack of informative prior knowledge – an uninformative prior.
>
> In response to the reviewer’s suggestion, we have empirically tested the impact of different constant values for these terms and included the results in the revised manuscript in Table 10 and 11 in the new Appendix B.3. For instance, the table at [this link](https://anonymous.4open.science/r/ICLR2024Rebutal-3528/prior_mean_vs_acc_table.png) (Table 10) evaluates different prior means, and [this link](https://anonymous.4open.science/r/ICLR2024Rebutal-3528/sigma_vs_acc.png) (Table 11) shows results for different prior diffusion constants. Our findings indicate that the prior mean has minimal effect, as all experiments converge to similar accuracy, while the prior diffusion constant plays a more significant role, with values below 5 yielding the best results. These findings, now included in the appendix, demonstrate that the model is robust to such variations and that our chosen constants do not compromise performance.
>
> The choice of a constant drift function of zero in the prior OU process reflects the standard uninformative prior commonly adopted in Bayesian frameworks. However, our framework is flexible and supports both constant and non-constant priors, similar to Gaussian Process models. For instance, one could define a prior with a drift function such as f(x, t) = 2x + t, or even utilize another neural network as the drift function, introducing time-dependent dynamics into the process. This flexibility enables the incorporation of more informative priors tailored to specific tasks, which could potentially enhance model performance. But this is beyond the point of our paper.
>
> Similarly, a constant diffusion term is standard practice (Calvo-Ordoñez et al. (2024); Xu et al. (2022); Li et al. (2020)). Parametrizing the diffusion with a learnable function (e.g., a neural network) often causes the diffusion term to collapse to zero, a well-documented issue in the literature (see, e.g., Li et al. (2020) or Xu et al. (2022), Section 2.1), since this would lead to the highest training accuracy. Furthermore, as noted in Calvo-Ordoñez et al. (2024), using the same diffusion function for both the prior and posterior is necessary to keep the KL divergence finite, further supporting the choice of a constant diffusion function.
>
> ---

---

> > ### Author Response · Authors · 2024-11-20
> > **Official Response to Reviewer 9vdg (Part 2)**
> >
> > ---
> >
> > > The experimental section would benefit from increased clarity and specificity, especially in regard to the structure and setup of experiments...
> >
> > We thank the reviewer for their suggestion. In response, we have included all details regarding the experimental setup, including the number of epochs and early stopping criteria, in [Table 17](https://anonymous.4open.science/r/ICLR2024Rebutal-3528/time_per_epoch.png) in Appendix B, with changes highlighted in blue in the updated submission. Below, we provide a brief summary:
> >
> > ➡️ Out-of-Distribution (OOD) Detection: Models were trained on all but one class, with the excluded class treated as OOD during validation and testing. Validation accuracy was recalculated on the adjusted split, and entropy thresholds were applied to evaluate OOD detection performance. See Section 5.2 for more details.
> >
> > ➡️ Noise Perturbation: Gaussian noise was added to the feature matrix in the test dataset only, and models were evaluated on the perturbed datasets. Details are provided in Section 5.3.
> >
> > ➡️ Active Learning (AL): For AL, models were iteratively trained by selecting the most uncertain nodes based on predictive entropy. Training continued with the newly acquired nodes until the training dataset was doubled. See Section 5.4 for additional information.
> >
> > We used an early stopping criterion based on validation accuracy, stopping training if no improvement was observed for 20 consecutive epochs. After performing a hyperparameter search ([details here](https://anonymous.4open.science/r/ICLR2024Rebutal-3528/grid_search.png) or Table 6 in the Appendix), the maximum number of epochs was set to 300.
> >
> > We hope these clarifications enhance reproducibility. If the reviewer has further questions or concerns about the experimental setup, please let us know so we can address them. Please note that all experiments will be fully reproducible when we release the code in the camera-ready version.
> >
> > ---
> >
> > > It is stated in Section 5 that hyperparameters that achieved the highest validation accuracy were chosen...
> >
> > We thank the reviewer for this valuable feedback. To improve clarity and reproducibility, we have included detailed information about the validation accuracy used and the grid search setup in Appendix B, with changes highlighted in blue in the revised submission.
> >
> > Regarding validation accuracy, we refer to the model's performance on the validation set using a standard dataset split (training-validation-test). The validation dataset was used to determine the epoch at which the model performed best, and the corresponding weights were used for evaluation on the test set.
> >
> > Details of the hyperparameter grid search can be found [here](https://anonymous.4open.science/r/ICLR2024Rebutal-3528/grid_search.png) and in Appendix B.1 of the revised manuscript. The selected hyperparameters were then applied consistently across all experimental setups.
> >
> > ---
> >
> > > Table 7 shows that uniform hyperparameters were chosen across all models...
> >
> > We appreciate the reviewer highlighting this point and apologize for the confusion. In fact, we did perform a hyperparameter search for each model independently. To reduce any misunderstanding, we have updated Table 7 to include only the parameters that were not part of our hyperparameter grid search. These changes are highlighted in blue in Appendix B.
> >
> > The grid search used for each model can be viewed here. The search included options for learning rate, weight decay, number of epochs, dropout rate, hidden dimensions, and step size. Each combination was evaluated, and the configuration yielding the highest validation accuracy was selected.
> >
> > We hope the hyperparameters and experimental setup are now clearer. Once again, this should be fully reproducible when we provide the code in the final version of the paper.
> >
> > ---

---

> > > ### Author Response · Authors · 2024-11-20
> > > **Official Response to Reviewer 9vdg (Part 3)**
> > >
> > > > The most impactful perceived weakness of this paper was the lack of experimental comparison with the models referenced as related works...
> > >
> > > We appreciate the reviewer’s valuable feedback and agree that including comparisons with the referenced methods is crucial. In the updated manuscript, we have added results for the state-of-the-art methods for uncertainty quantification in node classification, including GNSD (Lin et al., 2024) and GPN (Stadler et al., 2021), as suggested. Additionally, we have benchmarked further methods such as GKDE (Zhao et al., 2020), MSP (Hendrycks & Gimpel, 2016), GNNSafe (Wu et al., 2023), GRAND (Chamberlain et al., 2021), GREAD (Choi et al., 2023), Mahalanobis (Lee et al., 2018), and ODIN (Liang et al., 2018). The results are presented as additional columns in the experimental tables across all datasets and experiments, highlighted in blue. Summaries can also be viewed [here](https://anonymous.4open.science/r/ICLR2024Rebutal-3528/amazon_new_OOD.png) and [here](https://anonymous.4open.science/r/ICLR2024Rebutal-3528/cora_new_OOD.png) in the Appendix B.
> > >
> > > Our approach demonstrates competitive performance with GNSD while being simpler and easier to implement, and it outperforms all other methods for most datasets. We would also like to note that our existing baselines, including Graph GPs, BGCNs, and Ensembles, all leverage stochastic approaches, ensuring fair comparisons. We hope these additions address the reviewer’s concern and strengthen the paper.
> > >
> > > ---
> > >
> > > > For OOD detection, comparison against methods built specifically for this task on graphs, like GNNSafe (Wu et al., 2023) would be enlightening.
> > >
> > > We thank the reviewer for this important suggestion. In response, we have included the OOD detection results for the requested specialized method, GNNSafe (Wu et al., 2023), and GKDE (Zhao et al., 2020) in our updated manuscript, as indicated in our responses above (see [here](https://anonymous.4open.science/r/ICLR2024Rebutal-3528/amazon_new_OOD.png) and [here](https://anonymous.4open.science/r/ICLR2024Rebutal-3528/cora_new_OOD.png)). Note that this setup was taken directly from the paper from Lin et al., 2024.
> > >
> > > ---
> > >
> > > > One of the theoretical claims was a resistance to small perturbations in graph structure. This might lead to a compelling experiment, but this claim is not tested.
> > >
> > > We thank the reviewer for their comment. We would like to point out that the resistance to small perturbations in graph structure is essentially tested in Subsection 5.3 (Noise Perturbation), where our model consistently achieves top1 or top2 results. We have clarified this link between the experiment in 5.3 and Proposition 2 in the revised manuscript, with changes highlighted in blue in Subsection 5.3.
> > >
> > > ---
> > >
> > > > Is it possible to allow a non-constant drift function in the prior OU process? Would a similar method or extension allow this?
> > >
> > > We thank the reviewer for the question. Yes, it is possible to allow a non-constant drift function in the prior OU process, as our framework supports both constant and non-constant priors, similar to Gaussian Process models. For example, one could define a prior with a drift function such as $f(t \cdot 2, x) = -2t \cdot x$ or even use another neural network, introducing time-dependent dynamics into the process. This flexibility allows the incorporation of more informative priors tailored to specific tasks, potentially improving model performance.
> > >
> > > However, since an appropriate prior is not clear for the latent space in our datasets, we use an uninformative prior with a constant drift 0 function, which is standard practice in Bayesian ML (with a mean of 0). We would like to note that prior choice is a broad and active area of research within Bayesian methods, and a detailed exploration of this topic is beyond the scope of the current paper.
> > >
> > > ---
> > >
> > > > What is the effect upon training and convergence for different choices of hyperparameters...
> > >
> > > As discussed earlier, the table available at this [link](https://anonymous.4open.science/r/ICLR2024Rebutal-3528/prior_mean_vs_acc_table.png) evaluates various prior drifts, while [this link](https://anonymous.4open.science/r/ICLR2024Rebutal-3528/sigma_vs_acc.png) presents results for different prior diffusion constants. The findings indicate that variations in the prior drift do not significantly affect accuracy. Similarly, the prior diffusion constant (σ) has minimal impact on performance, provided it remains below a value of 5.
> > >
> > > ---

---

> > > > ### Author Response · Authors · 2024-11-20
> > > > **Official Response to Reviewer 9vdg (Part 4)**
> > > >
> > > > > How does the walltime of learning with an LGNSDE compare to walltime for the other tested methods?
> > > >
> > > > We thank the reviewer for this question. One disadvantage of our model is that it is slower than deterministic counterparts like GNODEs or standard Graph Neural Networks (e.g., GCNs). The additional computational cost arises from the numerical integration methods required for LGNSDEs. Specifically, sampling from the Brownian motion at each discretized during integration introduces a bottleneck, as seen when comparing Euler-Maruyama (used for stochastic systems) with standard Euler integration. However, this added complexity allows our model to capture both epistemic and aleatoric uncertainties, enabling more informative predictions.
> > > >
> > > > For reference, the average run times per epoch for each model on the Cora dataset can be found here: [link](https://anonymous.4open.science/r/ICLR2024Rebutal-3528/time_per_epoch.png).
> > > >
> > > > ---
> > > >
> > > > > Are there any interesting limitations on the practicality of this method, and in your experience, what can allow for learning the best results?
> > > >
> > > > On the one hand, the main limitation on the practicality of our method is the time complexity, see [complexity](https://anonymous.4open.science/r/ICLR2024Rebutal-3528/time_and_memory_complexity.png) and [components](https://anonymous.4open.science/r/ICLR2024Rebutal-3528/memory_and_time_varaible_names.png), or see Appendix C of the paper. This bottleneck is due to the SDE solver, which need multiple samples to estimate the true solution like in BNNs. However, these samples could be paralyzed in future work, helping with the time complexity. The memory complexity on the other hand, is one of the main advantages as it becomes constant $O(1)$ when employing the adjoint method (Chen et al., 2018).
> > > >
> > > > ---

---

> > > > > ### Author Response · Authors · 2024-11-20
> > > > >
> > > > > Thank you again for the review and all the comments. We would like to kindly ask if we have addressed all of your concerns. If so, could you increase your score accordingly? If there are any remaining concerns, please let us know so that we can address them promptly.

---

> > > > > > ### Author Response · Authors · 2024-11-25
> > > > > >
> > > > > > We kindly ask again if we have fully addressed all your concerns in our response. If so, would you consider increasing your score accordingly? Please note that the other two reviewers are favoring acceptance (10/10 and 6/10). Could you let us know if this aligns with your updated perspective?

---

> > > > > > > ### Comment · Reviewer_9vdg · 2024-11-27
> > > > > > > **Increased Score**
> > > > > > >
> > > > > > > Thank you for the thorough comments and responses! I have carefully reconsidered the work and raised my score. I do feel that many of my concerns were adequately addressed.

---

### Official Review · Reviewer_kdL1 · 2024-11-02

**Soundness:** 2
**Presentation:** 2
**Contribution:** 2
**Rating:** 6
**Confidence:** 3

**Summary:**

The paper proposes a new GNN architecture version based on stochastic differential equations to improve uncertainty estimation.

**Strengths:**

1. The paper demonstrates good empirical performance.
2. The authors evaluate uncertainty estimation through OOD detection, noise perturbation, and active learning.

**Weaknesses:**

1. The paper lacks comparisons with SOTA works. Some strong methods, such as GPN (Stadler et al., 2021) and GNSD (Lin et al., 2024), are mentioned but not evaluated, despite their better empirical performance compared to the baselines used. Additionally, other high-performing methods that utilize energy variants are not tested.
2. The method shows significant similarities to Lin et al. (2024), who also propose an SDE-based GNN. While the authors acknowledge the difference with one sentence in the related work section, I believe the similarities and differences should be further discussed in detail, such as by comparing the frameworks mathematically and conducting a specific study (even on a synthetic dataset) to highlight the unique merits of their approach. But the method is not compared against at all.

**Questions:**

1. Can you elaborate on how the 'framework effectively quantifies uncertainty' based on Proposition 1? It is not immediately clear to me how this bounded output variance translates to effectively quantified uncertainty.


While the topic and approach are interesting and the method appears potentially promising, I have concerns about the evaluation and the presentation, particularly the lack of differentiation from existing methods. If additional data and clarifications are provided, I would be willing to reconsider my rating.


---
Post rebuttal: experimental evaluation seems good now.

---

> ### Author Response · Authors · 2024-11-20
> **Official Response to Reviewer kdL1**
>
> Dear Reviewer kdL1, thank you for your detailed feedback and constructive comments. We are glad that you found the topic and approach interesting and appreciate your recognition of the model's potential and its evaluation of uncertainty estimation. We believe the updates and clarifications provided in our response and the revised manuscript directly address your concerns and demonstrate the unique strengths and contributions of our work. We hope these additions encourage you to reconsider your rating. Otherwise, please let us know so that we can address your concerns promptly.
>
> Now, find below our responses:
>
> ---
>
> > The paper lacks comparisons with SOTA works...
>
> We thank the reviewer for this helpful feedback. We agree that these comparisons are important and have added new experiments in the revised manuscript. Specifically, we now include results for state-of-the-art methods such as GNSD (Lin et al., 2024), GPN (Stadler et al., 2021), GKDE (Zhao et al., 2020), MSP (Hendrycks & Gimpel, 2016), GNNSafe (Wu et al., 2023), GRAND (Chamberlain et al., 2021), GREAD (Choi et al., 2023), Mahalanobis (Lee et al., 2018), and ODIN (Liang et al., 2018). GNNSafe, as an energy-based method, directly addresses the reviewer’s suggestion to include such approaches. The results are summarized in anonymized links: [Cora OOD](https://anonymous.4open.science/r/ICLR2024Rebutal-3528/cora_new_OOD.png) and [Amazon OOD](https://anonymous.4open.science/r/ICLR2024Rebutal-3528/amazon_new_OOD.png) which are table 16 and 17 in the Appendix.
>
> LGNSDE performed well in these comparisons, consistently ranking second-best across most metrics and achieving the best performance in some cases. For example, in the [Amazon OOD](https://anonymous.4open.science/r/ICLR2024Rebutal-3528/amazon_new_OOD.png) experiment, it achieved the highest AUPR (Out) score of 97.78. These additions provide a more comprehensive evaluation. We hope that we have addressed the reviewer’s concerns.
>
> ---
>
> > The method shows significant similarities to Lin et al. (2024), who also propose an SDE-based GNN...
>
> Thank you for the suggestion. We have now clarified the differences between LGNSDE and GNSD (Lin et al., 2024) in more detail in the paper. Specifically:
>
> GNSD  Lin et al. (2024) models message passing (information propagation in the graph) using a stochastic diffusion equation, similar to GRAND (Chamberlain et al., 2021), which uses an ODE for the same purpose. This approach integrates stochasticity directly into the graph diffusion process via a Q-Wiener process, enabling uncertainty propagation during node feature updates. Our model LGNSDE, in contrast, models the latent space of the graph as evolving stochastically under an SDE, while employing a standard GCN for message passing. Resulting in two fundamentally different approaches.
>
> As mentioned above, we have included the GNSD (Lin et al. (2024) and other baselines, see [Amazon OOD](https://anonymous.4open.science/r/ICLR2024Rebutal-3528/amazon_new_OOD.png), and [Cora OOD](https://anonymous.4open.science/r/ICLR2024Rebutal-3528/cora_new_OOD.png).
>
> ---
>
> > Can you elaborate on how the 'framework effectively quantifies uncertainty' based on Proposition 1?...
>
> We thank the reviewer for pointing this out. We revised the statement in the updated manuscript to better reflect our intended meaning and it now reads as follows: “... This ensures that the output variance is bounded by the prior variance of the latent space, providing a controlled measure of uncertainty”. This means that the variance of the output will not explode, since the variance has to be smaller or equal to the prior variance.
>
> Formally, the key point is that, in a Bayesian framework, the posterior variance $\text{Var}(\mathbf{H}(t))$ cannot exceed the prior variance. This ensures that the posterior is bounded by the prior, and as a result, the output variance $\text{Var}(\hat{\mathbf{y}}(t))$ is also bounded by the prior variance of the latent space through the relationship $\text{Var}(\hat{\mathbf{y}}(t)) \leq L_h^2 \text{Var}(\mathbf{H}(t))$ This property ensures that the uncertainty remains controlled and cannot grow arbitrarily large. We hope this clarification resolves the confusion and thank you again for the question.
>
> ---

---

> ### Author Response · Authors · 2024-11-20
>
> Thank you again for the review and all the comments. We would like to kindly ask if we have addressed all of your concerns. If so, could you increase your score accordingly? If there are any remaining concerns, please let us know so that we can address them promptly.

---

> > ### Comment · Reviewer_kdL1 · 2024-11-21
> >
> > Thanks for the rebuttal. The authors addressed most of my concerns, I have increased my score.

---

### Official Review · Reviewer_SZ1C · 2024-11-04

**Soundness:** 4
**Presentation:** 4
**Contribution:** 4
**Rating:** 10
**Confidence:** 3

**Summary:**

This paper generalizes GNODE to its stochastic counterpart, using SDEs to derive uncertainty-aware representations for graph-structured data. Theoretical and experimental characterization of the framework, named LGNSDE, showed its robustness and uncertainty quantification capability.

**Strengths:**

- The idea is sound and innovative.
- The mathematical framework is simply elegant.
- The robustness of this model is clearly demonstrated by theoretical and experimental results.
- This framework can be of high utility to the community.

**Weaknesses:**

- I would imagine the result is heavily dependent upon the integration methods. See the extensive study conducted in GRAND: Graph Neural Diffusion, Chamberlain et al. 2021.
- The accuracy of this model is not as performant as many cheaper variants.
- I would love to see its speed and memory benchmark, as I imagine it to be quite expensive.

**Questions:**

- Have you tried varying the backbones of the GNN and is there any performance change? I think rewiring the graph structure might have a huge impact.

---

> ### Author Response · Authors · 2024-11-20
> **Official Response to Reviewer SZ1C**
>
> We would like to start by thanking the reviewer for highlighting the strengths of our work, including the innovation of the idea, the elegance of the mathematical framework, and the potential demonstrated by our results. We hope that our detailed responses and the updates in the revised manuscript reaffirm your satisfaction with the paper. If any questions or concerns arise, please do not hesitate to let us know, and we will address them promptly.
>
> Please find below our responses to all your questions and comments ⤵️
>
> ---
>
> > I would imagine the result is heavily dependent upon the integration methods...
>
> We appreciate the observation made by the reviewer and agree with the claim. As noted in GRAND (Chamberlain et al., 2021), numerical integration methods play a significant role in the performance of diffusion-based models. Following their approach, we experimented with several numerical integration schemes, including SRK and Euler-Maruyama, and arrived at similar conclusions. Specifically, we found SRK to consistently perform the best, which aligns with GRAND’s findings (the stochastic variant of RK4) and is also well-supported in the literature as an effective method for solving stochastic differential equations (e.g., Kloeden & Platen, 1992; Milstein, 1995). Their results were a key motivator for our choice of SRK as the default method in our experiments.
>
> ---
>
> > The accuracy of this model is not as performant as many cheaper variants.
>
> We thank the reviewer for this comment. While the accuracy of our model is comparable to standard approaches, it does achieve slightly better results in most cases. However, the primary objective of our work was not to maximize accuracy but to develop a model capable of effectively quantifying uncertainty. In this regard, our model demonstrates strong performance in tasks such as out-of-distribution detection, noise perturbation handling, and active learning. The observed improvement in accuracy is an additional benefit but not the primary focus of our contributions.
>
> ---
>
> > I would love to see its speed and memory benchmark, as I imagine it to be quite expensive.
>
> For benchmarks of time and memory complexity, please refer to [complexity](https://anonymous.4open.science/r/ICLR2024Rebutal-3528/time_and_memory_complexity.png) and [components](https://anonymous.4open.science/r/ICLR2024Rebutal-3528/memory_and_time_varaible_names.png), or see Table 15 and 16 in the Appendix C of the paper. This bottleneck is due to the SDE solver, which need multiple samples to estimate the true solution just like BNNs. However, these samples could be parallelised in future work, helping with the time complexity. The memory complexity on the other hand, is one of the main advantages as it becomes constant $O(1)$ when employing the adjoint method (Chen et al., 2018).
>
> ---
>
> > Have you tried varying the backbones of the GNN and is there any performance change? I think rewiring the graph structure might have a huge impact.
>
> Thank you for the question. The choice of backbone is indeed important, and we have explored this aspect in our experiments. Specifically, we tested a Graph Attention Network (GAT) as a backbone instead of the GCN used in the original model. The results can be viewed [here](https://anonymous.4open.science/r/ICLR2024Rebutal-3528/GATvsGAT_cora_citseer.png) or Table 12 in the appendix C. We can see that  on the Cora dataset, LGNSDE-GAT outperformed LGNSDE-GCN, achieving an AUROC of 0.8249 compared to 0.7614, making it the best-performing variant so far. However, on the Citeseer dataset, LGNSDE-GAT underperformed relative to LGNSDE-GCN. For a comparison of the entropy distributions on the Cora dataset, see [this](https://anonymous.4open.science/r/ICLR2024Rebutal-3528/GATvsGCN_OOD_cora.png) figure (Figure 4 in the Appendix).
> The reviewer is right in noting that our proposed framework is flexible and allows the use of any graph model as a backbone, enabling adjustments to optimize performance depending on the dataset. Thank you again for the question.
>
> ---

---

> > ### Comment · Reviewer_SZ1C · 2024-11-27
> >
> > Thank you very much for your discussion!

---

### Author Response · Authors · 2024-11-20
**General Comments and Main Elements of our Rebuttal**

Dear reviewers, thank you for your constructive and thoughtful comments.

We are grateful for your feedback, which has helped us improve the clarity and scope of our work. We appreciate that **Reviewer kdL1** highlighted our strong empirical performance and evaluation of uncertainty estimation. **Reviewer 9vdg** recognized the clear model description, the originality of our theoretical results, and the potential for uncertainty quantification. **Reviewer SZ1C** commended the soundness and innovation of the idea, as well as the robustness demonstrated by the theoretical and experimental results.

**The main elements of our rebuttal are as follows:**

➡️ **Expanded Baseline Comparisons.** One major concern was the lack of sufficient baselines. We now include OOD detection comparisons with state-of-the-art methods such as GNSD (Lin et al., 2024), GPN (Stadler et al., 2021), GKDE (Zhao et al., 2020), MSP (Hendrycks & Gimpel, 2016), GNNSafe (Wu et al., 2023), GRAND (Chamberlain et al., 2021), GREAD (Choi et al., 2023), Mahalanobis (Lee et al., 2018), and ODIN (Liang et al., 2018). Our model consistently ranks among the top 1–3 across datasets. Updated results are provided [here (Amazon OOD)](https://anonymous.4open.science/r/ICLR2024Rebutal-3528/amazon_new_OOD.png) and [here (Cora OOD)](https://anonymous.4open.science/r/ICLR2024Rebutal-3528/cora_new_OOD.png), as well as in **Appendix B** of the updated manuscript. We hope these additional baselines address the reviewers’ concerns and highlight the effectiveness of our approach to quantifying uncertainty in graph data.

➡️ **Empirical Study of Prior Choices.** In response to comments on prior drift and diffusion functions, we conducted a study showing the impact of different choices. Results for [drift](https://anonymous.4open.science/r/ICLR2024Rebutal-3528/prior_mean_vs_acc_table.png) and [diffusion](https://anonymous.4open.science/r/ICLR2024Rebutal-3528/sigma_vs_acc.png), or in **Appendix B**, demonstrate robustness to drift constant variations and identify optimal ranges for diffusion constants.

➡️ **Improved Clarity and Reproducibility.** We have made clarifying edits throughout the manuscript to improve comparisons, theoretical explanations, and experimental setups. Additionally, all experiments will be fully reproducible in the final version with the provided code.

We hope these updates address your concerns and strengthen our submission. If you still have any concerns, please let us know so that we can address them. Please find detailed responses below and an updated version of our manuscript attached to the submission.

---

### Meta-Review · Area_Chair_niFy · 2024-12-20

**Metareview:**

This paper investigates uncertainty-aware representations in graph neural networks using stochastic differential equations (SDEs). The authors provide theoretical results for bounding the model output variance and bounding solutions under node feature perturbations. Experiments are conducted to validate the effectiveness of the proposed structure.

During the initial reviews, all reviewers acknowledged the novelty of the proposed method and the soundness of its theoretical results but raised concerns about the lack of empirical comparisons with state-of-the-art methods and insufficient discussion on the methodological differences with other SDE-based approaches. The authors successfully addressed these issues by including more comprehensive experiments and improving the writing.

While the performance of the proposed method is not particularly impressive compared to standard approaches, the reviewers agree that the method offers meaningful characterizations of uncertainty in graph data learning. Congratulations to the authors on this great work.

**Additional Comments On Reviewer Discussion:**

The reviewers asked for more empirical comparison with state-of-the-art methods. They also asked for clarification with other methods using SDEs. The authors successfully addressed these questions by providing more experiments and a clearer explaination on the methods.

---

### Decision · Program_Chairs · 2025-01-22

Accept (Spotlight)